# CATS: Mitigating Correlation Shift for Multivariate Time Series Classification

## Abstract

Unsupervised Domain Adaptation (UDA) leverages labeled source data to train models for unlabeled target data. Given the prevalence of multivariate time series (MTS) data across various domains, the UDA task for MTS classification has emerged as a critical challenge. In this work, we identify a key property of MTS: multivariate correlations vary significantly across domains, and further formalize this phenomenon as a novel type of domain shift, termed correlation shift. To mitigate correlation shift, we propose a scalable and parameter-efficient Correlation Adapter for MTS (CATS). Designed as a plug-and-play technique compatible with various Transformer variants, CATS employs temporal convolution to capture local temporal patterns and a graph attention module to model the changing multivariate correlation. The adapter reweights the target correlations to align the source correlations with a theoretically guaranteed precision. A correlation alignment loss is further proposed to mitigate correlation shift, bypassing the alignment challenge from the non-i.i.d. nature of MTS data. Extensive experiments on four real-world datasets demonstrate that (1) compared with vanilla Transformer-based models, CATS increases over 10% average accuracy while only adding around 1% parameters, and (2) all Transformer variants equipped with CATS either reach or surpass state-of-the-art baselines.

## 1 Introduction

Multivariate time series classification (MTS) is a fundamental task with applications spanning diverse fields, including finance (Zhao et al., 2023; Shahi et al., 2020; Mondal et al., 2014; LeBaron et al., 1999), healthcare (Zeger et al., 2006; Touloumi et al., 2004; Dockery & Pope, 1996; Bernal et al., 2017), climate science (Yoo & Oh, 2020; Ghil et al., 2002; Belda et al., 2014; Baranowski et al., 2015), transportation (Rezaei & Liu, 2019; MontazeriShatoori et al., 2020; Vu et al., 2018) and power systems (Hoffmann et al., 2020; Fütterer et al., 2017; Susto et al., 2018). Recently, deep learning models (Vaswani, 2017; Liu et al., 2023; Wu et al., 2022) have demonstrated remarkable capability in capturing temporal dependencies for MTS, showcasing significant promise in numerous applications.

However, the deployment of these models often encounters a critical challenge: *domain shifts* (Koh et al., 2021; Luo et al., 2019; Zhang et al., 2013) between the labeled source domain data during training and the target domain data during testing. The domain shift often leads to a notable degradation in model performance on the target domain. Moreover, obtaining labels for the test data is quite hard in most real-world scenarios (Ganin et al., 2016; Long et al., 2015). As a result, the UDA problem on MTS (He et al., 2023; Wilson et al., 2020) has emerged as a critical research area (He et al., 2023; Wilson et al., 2020), aiming to leverage the labeled source domain data to enhance the model performance on the unlabeled target domain.

Previous studies on UDA for MTS primarily focus on learning domain-invariant features through adversarial training (Wilson et al., 2020; 2023), contrastive learning (Ozyurt et al., 2022) or divergence metrics (He et al., 2023; Cai et al., 2021). However, these approaches have two notable limitations: (1) Model architecture perspective: **existing UDA methods exhibit limited adaptability across model architectures.** Most prior approaches are tightly coupled with specific backbone networks, typically built on lightweight RNNs or CNNs (Liu & Xue, 2021; Wilson et al., 2020; He et al., 2023; Li et al., 2022b; Wilson & Cook, 2020). This

coupling limits their scalability and applicability to more expressive and modern MTS analysis models, such as iTransformer (Liu et al., 2023) and Crossformer (Zhang & Yan, 2023). Compared with existing UDA backbone networks, these models are typically Transformer variants[1] with larger parameter capacities and are better suited for modeling rich and complex temporal patterns in real-world time series. However, these architectures are not well supported by existing UDA frameworks. This gap motivates the key question: *Can we develop model-agnostic UDA methods that effectively enhance the adaptability of powerful general MTS models?* (2) Data distribution perspective: **Prior works lack architectural designs for addressing the variance of multivariate correlations across domains.** We observe consistent and significant shifts in inter-variable dependencies, i.e., correlations, across domains, and we term this new type of domain shift as **correlation shift**. Although a few existing studies (Wang et al., 2024a; Cai et al., 2021) implicitly acknowledge this issue, their primary focus has been on designing loss functions for alignment, with little discussion on how architectural modules can be tailored to fundamentally mitigate correlation shift. This oversight in model design leaves significant room for improvement on this challenge.

To overcome these limitations, we seek to align the correlation distributions between domains from both the model architecture and the training objective perspectives. At the model level, we propose a scalable and parameter-efficient adapter, termed Correlation Adapter for Multivariate Time Series (CATS). Specifically, to capture temporal dependencies, CATS employs depthwise convolutions along the temporal dimension for both down-projection and up-projection. Compared to traditional adapters that rely on linear matrices, convolutions demonstrate superior capability in capturing local temporal patterns. Building on this, CATS incorporates a Graph Attention Network (GAT) to adaptively reweight inter-variable dependencies in the hidden layers. Theoretically, proper reweighting can align the correlation distributions across domains, thus mitigating correlation shift. To adapt CATS to the unlabeled target domain, we introduce a novel correlation alignment loss. This loss function not only effectively reduces correlation shift but also circumvents the limitations of divergence metrics, which often fail on the non-i.i.d. (non-independent and identically distributed) time series data. By integrating CATS with this tailored loss function, we present a more effective and efficient solution for unsupervised domain adaptation in multivariate time series.

In summary, our main contributions are as follows:

- **Problem.** We are the first to empirically validate and mathematically define the concept of correlation shift in MTS data. As a unique and important property of MTS data, correlation shift introduces a distinct perspective for addressing UDA challenges.
- **Model.** We propose the first scalable and parameter-efficient MTS adapter, CATS, designed to be highly adaptable across different model architectures. Empirically and theoretically, CATS effectively mitigates correlation shifts while capturing local temporal patterns for classification.
- **Training objective.** We introduce a novel correlation alignment loss, which directly addresses correlation shift and circumvents the alignment challenge posed by the non-i.i.d. MTS data.
- **Evaluation.** We conduct extensive experiments on four real-world time series domain adaptation datasets. The results demonstrate that CATS consistently enhances the performance of MTS analysis models, achieving a 10%+ average accuracy improvement, even under large domain shifts. Furthermore, MTS analysis models equipped with CATS outperform SOTA baselines by around 4% average accuracy, showcasing the superb effectiveness and adaptability of CATS.

## 2 Preliminaries

**Multivariate time series classification.** In the task of multivariate time series (MTS) classification, the dataset is comprised of a collection of time series samples along with their corresponding labels, denoted as $\mathcal{D} = \{(\mathbf{X}_i, y_i)\}_{i=1}^{n}$ with $n$ being the sample number. Here, the $i$-th sample $\mathbf{X}_i \in \mathbb{R}^{D \times T}$ represents an individual time series that contains readouts of $D$ observations over $T$ time points, and $y_i$ is the associated label. In this paper, we use $\mathbf{X}[j]$ to represent the $j$-th variable of the sample $\mathbf{X}$.

---

[1]As general MTS models are mostly Transformer-based, our discussion in this paper mainly focuses on Transformer variants. However, CATS can be easily extended to other block-wise non-Transformer architectures like TimesNet.

**Adapters for large models.** Recently, large-scale Transformers have achieved great success in various domains, including natural language processing (Vaswani, 2017; Devlin, 2018; Brown et al., 2020), computer vision (Radford et al., 2021; Alexey, 2020), and time series analysis (Liu et al., 2023; Wu et al., 2022). However, their large parameter size makes it impractical to fine-tune a separate model for every downstream task. To overcome this, a variety of parameter-efficient fine-tuning (PEFT) methods have been proposed (Han et al., 2024; Hu et al., 2021; Xu et al., 2023). Among them, *adapters* have attracted particular interest for their ability to transfer the knowledge of pretrained models to new tasks using only a small number of additional parameters.

Given the high similarity between the objectives of adapters and UDA, many studies (Zhang et al., 2021; Malik et al., 2023) have leveraged adapters to transfer knowledge learned from the source domain to the target domain. Such domain adapters are embedded between two consecutive Transformer blocks to adapt the model's learned representations to the target domain. Mathematically, these adapters can be expressed as:

$$\mathbf{H}_O^{(k)} = \mathbf{H}_I^{(k)} + \sigma(\mathbf{H}_I^{(k)}\mathbf{W}_\downarrow^{(k)})\mathbf{W}_\uparrow^{(k)} \tag{1}$$

where $\sigma(\cdot)$ represents the activation function, and $\mathbf{W}_\downarrow^{(k)} \in \mathbb{R}^{T \times r}$ and $\mathbf{W}_\uparrow^{(k)} \in \mathbb{R}^{r \times T}$ are the two linear matrices for down-projection and up-projection with $r$ being a small hyperparameter. Here, $\mathbf{H}_I^{(k)}$ is the input of the $k$-th adapter block, and $\mathbf{H}_O^{(k)}$ is the output of the $k$-th adapter, i.e., the input of the $(k+1)$-th Transformer block.

**Unsupervised domain adaptation.** The goal of UDA is to leverage information from a labeled source domain $\mathcal{D}_s = \{(\mathbf{X}_{i,s}, y_{i,s})\}_{i=1}^{n_s}$ to enhance the model's understanding of an unlabeled target domain $\mathcal{D}_t = \{\mathbf{X}_{i,t}\}_{i=1}^{n_t}$. Generally, source and target samples are independently sampled from their respective distributions, i.e., $\mathcal{D}_s \sim \mathcal{P}_s(\mathbf{X}, y)$ and $\mathcal{D}_t \sim \mathcal{P}_t(\mathbf{X})$. Here, $\mathcal{P}_s(\mathbf{X}, y)$ denotes a joint distribution in the source domain and $\mathcal{P}_t(\mathbf{X})$ denotes the marginal counterpart in the target domain. However, these distributions often exhibit significant shifts. There are two widely studied shifts: feature shift (Zhang et al., 2013) and label shift (Azizzadenesheli et al., 2019). Specifically, feature shift occurs when the distribution of features changes across domains, while the relationship between features and labels remains consistent. In contrast, label shift arises when the label distributions differ between domains, even if the feature distributions are similar.

## 3 Correlation Shift

Although label shift and feature shift are the two most commonly analyzed types of domain shifts in UDA tasks, focusing solely on these shifts is insufficient for MTS classification. A key characteristic of MTS is the interaction between different variables, such as the interplay between blood glucose levels and insulin in the human body (Basu et al., 2009; Wang et al., 2018). Correlation effectively models this inter-variable dependencies, thus making it central to many statistical and deep learning models for MTS (Box et al., 2015; Bollerslev, 1990; Wu et al., 2021).

Despite its importance, to our best understanding, no prior work has provided systematic research on the correlation distribution across domains for MTS data. To bridge this gap, we introduce a novel domain shift tailored specifically for MTS: correlation shift.

**Definition 1** (Correlation shift). *Suppose the source multivariate data* $\mathbf{X}_s \in \mathbb{R}^{D \times T}$ *and the target multivaraite data* $\mathbf{X}_t \in \mathbb{R}^{D \times T}$ *follow the source distribution* $\mathcal{P}_s(\mathbf{X})$ *and the target distribution* $\mathcal{P}_t(\mathbf{X})$, *respectively. Here, $D$ denotes the number of variables and $T$ represents the feature dimension. Then, **correlation shift** occurs when the multivariate correlations between the source and target domains differ, formally defined as:*

$$\mathrm{Corr}(\mathbf{X}_s) \neq \mathrm{Corr}(\mathbf{X}_t) \tag{2}$$

*where the correlation structure* $\mathrm{Corr}(\cdot)$ *is given by:*

$$
\begin{aligned}
\mathrm{Corr}(\mathbf{X}) &:= \mathrm{diag}(\mathbf{\Sigma})^{-1/2}\mathbf{\Sigma}\,\mathrm{diag}(\mathbf{\Sigma})^{-1/2}\\
\mathbf{\Sigma} &= \mathbb{E}_{\mathbf{X}\sim\mathcal{P}}\left[(\mathbf{X} - \mathbb{E}\mathbf{X})(\mathbf{X} - \mathbb{E}\mathbf{X})^T\right]
\end{aligned}
\tag{3}
$$

This phenomenon naturally arises from discrepancies in inter-variable dependencies across domains. A practical example of the correlation shift can be observed in healthcare analytics. For example, in non-diabetic individuals, there is typically a synchronous peak in blood glucose and insulin levels following sugar intake while in diabetic patients, the increase in insulin occurs with a noticeable delay after the peak in blood glucose (Basu et al., 2009; Wang et al., 2018). This delay represents a clear correlation shift when considering blood glucose and sugar intake as two interacting variables. Another widely-existing example comes from the weather data. Extensive studies (Draper & Long, 2004; Weissman et al., 2002; Back & Bretherton, 2005) have shown that the relationship between wind speed and precipitation varies geographically and this relationship tends to be significantly stronger in humid regions compared to arid areas. The widespread occurrence of correlation shifts impacts the transferability of learned representations, ultimately leading to deteriorated performance on target domains

To further validate the universality of correlation shifts, we conduct an empirical analysis on a real-world Human Activity Recognition (HAR) dataset (Anguita et al., 2013), to demonstrate the discrepancy in the correlation across different domains. Specifically, we iterate through the 30 domains in HAR, treating each domain as the source domain while considering the remaining 29 domains as target domains. For each source-target domain pair, we apply the Mann-Whitney U test (McKnight & Najab, 2010), a non-parametric hypothesis testing method, to determine whether there is a significant correlation shift between the source and the target domains. A detailed explanation for this correlation shift test is provided in Appendix A. We compute the rate of target domains suffering from significant correlation shifts for each source domain, and the results are shown in Figure 1, where the orange bars indicate the rate of target domains with significant correlation shifts. The red dashed line in Figure 1 marks the average rate of correlation shift, which is 78%. These findings provide clear evidence that correlation shifts are indeed prevalent in MTS datasets, thereby calling for solutions to mitigate them.

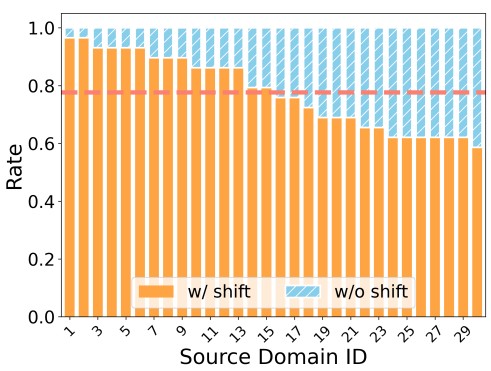

Figure 1: Rates of target domains with correlation shifts per source domain. The x-axis represents the source domain index while the y-axis indicates the rate of correlation shifts among the rest 29 domains. The red line marks the average rate of 78%.

## 4 Methodology

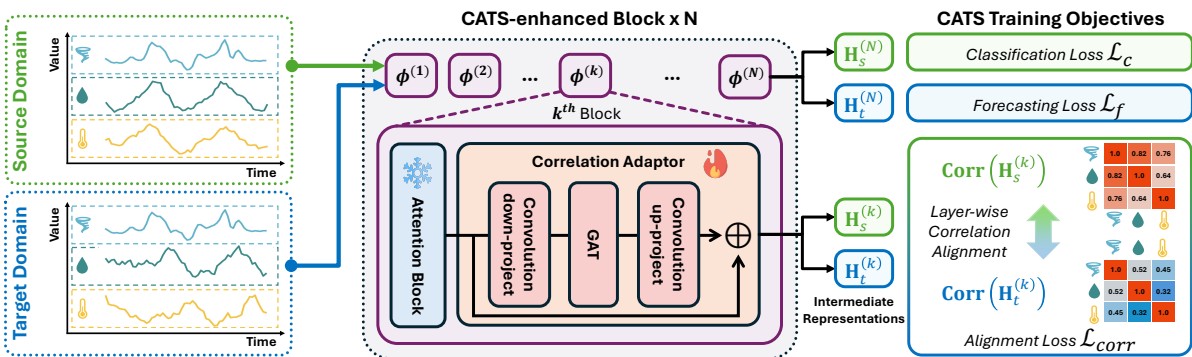

Figure 2: The main framework of CATS. CATS is integrated after each attention block of any Transformer variant, with only CATS trained and the backbone frozen. The training objective involves three loss functions: (1) classification loss on the labeled source domain, (2) forecasting loss on the unlabeled target domain, and (3) layer-wise correlation alignment loss to align these two domains.

In this section, we introduce our solution to mitigate correlation shift. We first propose CATS in Section 4.1, which demonstrates superior representation learning capabilities for MTS compared to traditional adapters.

By reweighting multivariate correlation, CATS enjoys theoretical guarantees for mitigating correlation shift. In Section 4.2, we propose a novel training objective for CATS on unlabeled target domains, centered on the correlation alignment loss to address correlation shift effectively. The overall framework of CATS is visualized in Figure 2.

### 4.1 CATS: Temporal-aware Correlation Adapter

MTS data exhibit two prominent properties: temporal dependencies and inter-variable dependencies. To model these properties, we first introduce up-project and down-project modules for time series adapters in Section 4.1.1 to capture temporal dependencies effectively. In Section 4.1.2, we further propose a reweighting module to adaptively refine inter-variable dependencies, thereby mitigating the impact of correlation shift. By integrating these components, we propose CATS which effectively enhances the model's adaptability in domain adaptation tasks involving multivariate time series classification.

### 4.1.1 Temporal Project via Convolution

As discussed in Section 2, adapters hold significant potential for addressing domain adaptation challenges in Transformer models. However, these previous adapters often rely on the assumption that the data are i.i.d. (independent and identically distributed), which does not hold for MTS. A key property of MTS is that temporally adjacent data points often exhibit strong similarity. However, the use of linear matrices in existing adapters fails to capture this local similarity, leading to noticeable declines in performance. Inspired by temporal convolution network (TCN) (Fan et al., 2023; Farha & Gall, 2019; Hewage et al., 2020), we posit that convolutions on temporal dimension better leverage local similarity on MTS, thus serving as a better substitute as project layer, compared with linear matrices in adapters from Eq. (1). Specifically, given a input representation of the adapter $\mathbf{H} \in \mathbb{R}^{D \times T}$, this temporal convolution can be formulated as follows:

$$\text{CONV}(\mathbf{H})[:, j] = \sum_{p=0}^{r-1} \mathbf{K}[p] \mathbf{H}[:, j-p] \tag{4}$$

where $\text{CONV}(\mathbf{H}) \in \mathbb{R}^{D \times T}$ denotes the output representation of the adapter, and $\mathbf{K} \in \mathbb{R}^{r \times D \times D}$ represents the convolution kernel with its kernel size of $r$.

However, one potential drawback of using convolutions along the temporal dimension is the increase of the number of trainable parameters. On previous adapters in Eq. (1), the parameter complexity of the linear matrices are $\mathcal{O}(T \times r)$. In contrast, convolutions have a parameter complexity of $\mathcal{O}(D^2 \times r)$. When the hidden layer dimension $D$ approaches or exceeds the time length $T$, the number of trainable parameters for convolutions can become quite large. To address this issue, we adopt depthwise convolutions (Chollet, 2017), where each variable is convolved with its own kernel. Mathematically, the temporal depth convolutions (TDC) can be defined as:

$$\text{TDC}(\mathbf{H})[i, j] = \sum_{p=0}^{r-1} \mathbf{k_i}[p] \mathbf{H}[i, j-p] \tag{5}$$

where $\mathbf{k_i} \in \mathbb{R}^r$ represents the TDC kernel on the $i$-th variable. This approach reduces the parameter complexity to $\mathcal{O}(D \times r)$, significantly improving efficiency. Note that depthwise convolutions ignore multivariate correlations. We will address this issue in Section 4.1.2.

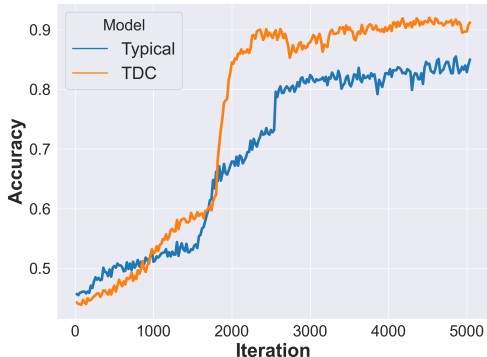

Figure 3: The accuracy comparison on HAR dataset between the typical adapter and the TDC-based adapter. With the backbone (TimesNet) pretrained on the domain 1, both adapters are trained on the domain 10.

To empirically validate the superiority of temporal depthwise convolutions, we compare the performance between a typical adapter in Eq. (1) and a TDC-based adapter which only uses temporal depthwise

convolutions instead of the linear layers. Specifically, given a backbone pretrained on the source dataset, we will train these two adapters on the target domain. Note that the task is designed to assess the representation learning ability of different adapters on time series rather than focusing solely on domain adaptation. Therefore, we use the labels from the target domain when training adapters. A detailed experimental setting is provided in Appendix B. The experiment results, illustrated in Figure 3, indicate that the TDC-based adapter demonstrates significantly higher accuracy (around 10% improvement) after both adapters converge. This clearly supports the premise that, for time series data, temporal convolutions are a superior alternative to traditional linear matrices.

### 4.1.2 Correlation Alignment via Reweighting

In this section, our objective is to identify an effective reweighting module to mitigate the correlation shift. Since correlation shift arises from discrepancies in multivariate correlations between the source and target domains, a natural approach to addressing it is to adaptively reweight the correlations in the target domain. Interestingly, in the case of Gaussian variables, we prove that a simple linear mapping[2] is sufficient to serve as an optimal reweighting function to align not only the correlation but also the joint distribution between variables. Mathematically, this finding could be formalized as Proposition 1. All proofs in this section are postponed to Appendix C.

**Proposition 1** (Gaussian Probability Alignment). *Suppose source data $\mathbf{X}_s \in \mathbb{R}^{D \times T}$ and target data $\mathbf{X}_t \in \mathbb{R}^{D \times T}$ follow $\mathcal{N}(\boldsymbol{\mu}_s, \boldsymbol{\Sigma}_s)$ and $\mathcal{N}(\boldsymbol{\mu}_t, \boldsymbol{\Sigma}_t)$, respectively. There exists a reweighting matrix $\mathbf{A}^\star \in \mathbb{R}^{D \times D}$ and a bias vector $\mathbf{b} \in \mathbb{R}^D$, such that the multivariate joint probability of the reweighted target domain perfectly aligns with that of the source domain, that is for every $i, j = 1, 2, ..., D$*

$$\Pr\left(\mathbf{X}_s[i], \mathbf{X}_s[j]\right) = \Pr\left(\mathbf{Y}[i], \mathbf{Y}[j]\right)$$

*where $\mathbf{Y} = \mathbf{A}^\star \mathbf{X}_t + \mathbf{b}$ and $\mathbf{b} = \mathbf{0}$ for most MTS data.*

This insight offers a promising direction for designing effective solutions without resorting to complex and computationally expensive methods. Importantly, even when the distributions of random variables are complex and difficult to characterize precisely, this simple linear mapping approach can still perfectly match the correlation between variables, thereby effectively mitigating correlation shifts, as shown in the following proposition.

**Proposition 2** (Correlation Alignment). *Suppose source data $\mathbf{X}_s \in \mathbb{R}^{D \times T}$ and target data $\mathbf{X}_t \in \mathbb{R}^{D \times T}$ follow the source distribution $\mathcal{P}_s(\mathbf{X})$ and the target distribution $\mathcal{P}_t(\mathbf{X})$, respectively. There exists a reweighting matrix $\mathbf{A}^\star \in \mathbb{R}^{D \times D}$, such that the correlation of the source distribution and target distribution can be perfectly aligned, formally expressed as:*

$$\text{Corr}(\mathbf{X}_s) = \text{Corr}(\mathbf{A}^\star \mathbf{X}_t) \tag{6}$$

Intuitively, the interaction between variables can be considered as a fully-connected graph, where each node represents a variable and edges model the inter-variable dependency, and the reweighting matrix $\mathbf{A}^\star$ serves as the adjacency matrix of the graph. However, in practical scenarios, solving the reweighting matrix $\mathbf{A}^\star$ is often computationally expensive and non-trivial, particularly when the distributions of the variables are highly complex. To address this, we leverage a Graph Attention Network (GAT) (Velickovic et al., 2017) to adaptively approximate the matrix $\mathbf{A}^\star$. Mathematically, we formalize this insight through the following theorem:

**Theorem 1** (Attention Approximation). *Let $\mathbf{A}^\star$ denote the optimal reweighting matrix defined in Proposition 2. Suppose the input matrix $\mathbf{X} \in \mathbb{R}^{D \times T}$ is drawn from a high-dimensional distribution with probability measure $P$. Assume that the following conditions hold:(1) $\mathbf{X}$ has full column rank;(2) The covariance matrix of $\mathbf{X}$ exists and possesses strictly distinct eigenvalues.Under these conditions, there exists a one-layer Graph Attention Network with a hidden dimension of $D$ such that*

$$\mathbb{E}_P\left[\text{MSE}\left(\text{GAT}\left(\mathbf{X}\right) - \mathbf{A}^\star \mathbf{X}\right)\right] \leq \frac{C}{D} \tag{7}$$

---

[2]Although adapters in Eq. (1) contains linear matrices, it is not a linear mapping due to the existence of the activation function.

*where $C > 0$ are a constant, and GAT follows the following formula:*

$$\text{GAT}(\mathbf{X}) = \mathbf{A}\mathbf{X}\mathbf{W}, \quad \mathbf{A}[i,j] = \sigma\left(\mathbf{a}^\top\left[\mathbf{W}^T\mathbf{X}[i] \,\|\, \mathbf{W}^T\mathbf{X}[j]\right]\right) \tag{8}$$

*where $\sigma : \mathbb{R} \to \mathbb{R}$ is a non-linear activation function, $\mathbf{A}[i,j]$ denote the $(i,j)$-th entry of $\mathbf{A}$, $\mathbf{W} \in \mathbb{R}^{T \times T}$ is a learnable linear projection. $\mathbf{a} \in \mathbb{R}^{2T}$ is a learnable attention vector, and $\|$ denotes vector concatenation. Consequently, as the input dimension $D$ approaches infinity, the GAT output converges to the optimal linear transformation $\mathbf{A}^\star\mathbf{X}$ in the mean squared sense:*

$$\lim_{D \to \infty} \mathbb{E}_P\left[\text{MSE}\left(\text{GAT}(\mathbf{X}) - \mathbf{A}^\star\mathbf{X}\right)\right] = 0.$$

The detailed proof of Theorem 1 is provided in Appendix C.3. By integrating the GAT and TDCs, we finally propose CATS to both well capture temporal dependencies and solve correlation shift. Mathematically, the CATS layer $\phi(\cdot)$ could be expressed as:

$$\phi(\mathbf{X}) = \mathbf{X} + \text{TDC}_\uparrow\left(\sigma(\text{GAT}\left(\text{TDC}_\downarrow\left(\mathbf{X}\right)\right))\right) \tag{9}$$

where $\text{TDC}_\downarrow$ and $\text{TDC}_\uparrow$ represent the temporal convolution layers for down-project and up-project, respectively. Here, GAT represents one GAT layer on a fully connected graph. We prove in Appendix C.4 that CATS in Eq. (9) also approximate the reweighting matrix $\mathbf{A}$ in a manner similar to a one-layer GAT.

Similar to typical domain adapters in Eq. (1), we integrate CATS within two consecutive blocks of a Transformer-based model. Since Transformers always consist of multiple blocks, this allows each instance of CATS to make minor adjustments to the target domain's distribution. Cumulatively, these incremental adjustments are capable of mitigating large domain shift from the final block. The idea of gradually reducing domain shift is conceptually similar to but bears subtle difference from gradual domain adaptation (He et al., 2024). Gradual domain adaptation leverages intermediate domains for supervision, which are often predefined. In contrast, CATS does not rely on any intermediate domain, making it a more flexible and efficient solution for UDA.

### 4.2 Training Procedures

In this section, we aim to propose an effective unsupervised training strategy for CATS. Our training strategy is designed to meet three critical objectives: (1) enable the model to effectively extract features from target domain samples; (2) maintain strong classification capabilities; and (3) align feature distributions between the source and target domains. To meet these objectives, we introduce three distinct loss functions that collectively guide the effective training of CATS.

First, to improve the model's understanding of the target domain, prior UDA works (He et al., 2023; Ghifary et al., 2016; Zhuo et al., 2017) often rely on reconstruction loss, which serves as an additional supervision for the classification on unlabeled target domain. Reconstruction loss ensures that the decoded output closely resembles the input on the target domain, requiring the encoded features to preserve all information from the target domain. However, the usage of reconstruction loss could be harmful to MTS classification. In MTS classification tasks, temporal properties, such as periodicity and trends, are more strongly correlated with the labels whereas local noise may be detrimental to classification performance. But reconstruction loss does not differentiate between meaningful features and noise, making it a suboptimal choice for such tasks. To address this issue, we propose the use of forecasting loss as an alternative. Forecasting tasks inherently require the model to focus more on temporal properties like trends and periodicity, while ignoring local random fluctuations. Consequently, features extracted for forecasting naturally transfer well to the classification task.

However, directly applying the forecasting loss presents challenges since the forecasting task requires the time series data to be sliced into adjacent historical and future segments. To address this, given a sample $\mathbf{X}_i^t \in \mathbb{R}^{D \times T}$ from the target domain, we first slice the samples into overlapping time windows $\mathcal{W}_{i,k}^t = \mathbf{X}_i^t[:, k : k + L], k \in \{1, \ldots, T - L\}$ [3] with $L$ being the window length, and then use those sliced time windows as the

---

[3] We use $\mathbf{X}[:, t_1 : t_2]$ to represent a sliced segment from time $t_1$ to time $t_2$ of $\mathbf{X}$. All the slicing notations follow Python standards.

training inputs. Specifically, given a model integrated with CATS, we leverage all the blocks without the last classification head as the feature extractor $f_{\texttt{CATS}}(\cdot)$ and introduce a new forecasting projection head $g_{\texttt{f}}(\cdot)$. Then the forecasting loss could be represented as:

$$\mathcal{L}_{\texttt{f}} = \sum_{i=1}^{n_t} \sum_{k=1}^{T-2L} \text{MAE}(g_{\texttt{f}}(f_{\texttt{CATS}}(\mathcal{W}_{i,k}^t)), \mathcal{W}_{i,k+L}^t) \tag{10}$$

where MAE represent the mean absolute error. Here, $\mathcal{W}_{i,k}$ and $\mathcal{W}_{i,k+L}$ are two adjacent time windows, indicating the history information and future, respectively.

Second, to ensure that CATS maintains the classification capabilities of the pretrained model, we perform a classification task using the labeled data from the source domain. To make the temporal dimension align with the previous forecasting task, we calculate the cross-entropy loss for each sliced time window:

$$\mathcal{L}_{\texttt{c}} = \sum_{i=1}^{n_s} \sum_{k=1}^{T-L} \ell_{\text{CE}}(g_{\texttt{c}}(f_{\texttt{CATS}}(\mathcal{W}_{i,k}^s)), y_i^s) \tag{11}$$

where $\ell_{\text{CE}}$ is the cross-entropy loss function, $g_{\texttt{c}}$ is the classification head, and $\mathcal{W}_{i,k}^s$ is the sliced time window from the source domain. During the inference phase, we randomly sample $m$ time windows from the entire time series and use a majority voting scheme to predict the label for the entire sequence based on the predictions from the sliced windows.

Third, to mitigate distribution shifts across domains, we propose the correlation alignment loss, specifically designed to address correlation shift. Given CATS's ability to adaptively reweight multivariate correlation, as discussed in Section 4.1.2, our objective is to align the correlation distribution of the source domain with that of the target domain transformed by CATS. Specifically, given the output of the $k$-th block $\mathbf{H}_s^{(k)}$ and $\mathbf{H}_t^{(k)}$ from the source domain and the target domain respectively, we minimize the Maximum Mean Discrepancy (MMD) (Gretton et al., 2012) of the correlation distribution between $\mathbf{H}_s^{(k)}$ and $\phi(\mathbf{H}_t^{(k)})$. Mathematically, the correlation alignment loss can be expressed as:

$$\mathcal{L}_{\textbf{corr}} = \sum_{k=1}^{K} \sum_{\substack{\mathbf{H}_s^{(k)} \\ \mathbf{H}_t^{(k)}}} \text{MMD}\left( \text{corr}\left( \mathbf{H}_s^{(k)} \right), \text{corr}\left( \phi^{(k)}\left( \mathbf{H}_t^{(k)} \right) \right) \right)$$

where $\text{corr}(\mathbf{H}) = \text{vec}\left( \frac{\mathbf{H}\mathbf{H}^T}{\|\mathbf{H}\|_{\mathcal{F}}^2} \right)$, $\text{MMD}(\cdot, \cdot)$ denotes the MMD loss, $\text{vec}(\cdot)$ denotes the vectorization operator and $\phi^{(k)}(\cdot)$ represent our CATS adapter in the $k$-th block defined in Eq. (9).

Compared to directly aligning hidden features using MMD, the correlation alignment loss offers a unique advantage in terms of optimization difficulties. This is primarily because MMD assumes that data distributions are i.i.d., while time series data inherently exhibit non-i.i.d. characteristics. Consequently, directly applying MMD to align feature distributions often increases the difficulty of optimization, potentially leading to suboptimal performance. In contrast, correlation alignment loss focuses on aligning correlations rather than directly aligning raw features. Within the same domain, these correlations across variables tend to be more stable compared to the feature distributions. For instance, in a financial time series dataset, the correlation between stock prices of two closely related companies might remain consistent over time, even though the individual stock price values fluctuate significantly (Kim & Baginski, 2016). Thus, within a single domain, if we consider the correlation of MTS data as a new "feature", this "feature" tends to exhibit higher similarity across different samples. Consequently, using MMD to align correlations becomes less challenging in terms of optimization. Therefore, this property makes correlation alignment loss a more stable and effective approach for reducing distributional discrepancies, particularly in MTS tasks, where temporal dependencies and multivariate correlation play a crucial role

To sum up, we combine the classification loss $\mathcal{L}_{\texttt{c}}$ on the source domain, the forecasting loss $\mathcal{L}_{\texttt{f}}$ on the target domain, and the correlation alignment loss $\mathcal{L}_{\textbf{corr}}$ across these two domains to formulate the final loss function. This unified objective ensures that the model not only learns discriminative features for classification but also

Table 1: Main accuracy results for MTS classification on the UDA task. The higher the accuracy is, the better. For three Transformer variants, the columns of 'w/o CATS', 'w/ CATS' and 'Δ' represent the accuracy without CATS, the accuracy with CATS, and the accuracy improvement due to CATS. **Bold font** indicates the best performance across **all the methods**, and underline symbol represents the best performance among UDA baselines.

| Dataset | UDA Baseline | | | | | | Transformer | | | TimesNet | | | Crossformer | | | iTransformer | | |
|---|---|---|---|---|---|---|---|---|---|---|---|---|---|---|---|---|---|---|
| Source → Target | CORAL | CoDATS | Raincoat | CLUDA | SASA | UDApter | w/o CATS | w/ CATS | Δ | w/o CATS | w/ CATS | Δ | w/o CATS | w/ CATS | Δ | w/o CATS | w/ CATS | Δ |
| HAR 24 → 27 | 78.76 | 82.30 | 96.88 | 82.14 | 86.72 | 96.46 | 91.81 | 98.23 | 6.42 | 93.69 | 97.34 | 3.65 | 80.53 | 93.80 | 13.27 | 82.30 | **99.11** | 16.81 |
| HAR 3 → 13 | 63.63 | 52.52 | 91.67 | 77.55 | 78.78 | 90.90 | 79.79 | **98.98** | 19.19 | 84.96 | 87.86 | 2.90 | 84.84 | 96.96 | 12.12 | 75.75 | 97.97 | 22.22 |
| HAR 16 → 13 | 47.47 | 39.40 | 71.87 | 69.39 | 61.61 | 66.67 | 73.96 | 77.78 | 3.82 | 67.67 | 83.84 | 16.17 | 69.69 | 87.87 | 18.18 | 74.74 | **85.86** | 11.12 |
| HAR 3 → 8 | 51.76 | 71.76 | 78.13 | 78.57 | 64.70 | 71.76 | 54.11 | 75.12 | 21.01 | 64.70 | **92.92** | 28.22 | 61.17 | 62.35 | 1.18 | 74.11 | 91.77 | 17.66 |
| HAR 19 → 2 | 61.53 | 60.00 | 76.56 | 60.00 | 69.23 | 59.34 | 53.84 | 73.52 | 19.68 | 53.84 | 82.41 | 28.57 | 48.35 | 53.84 | 5.49 | 59.34 | **84.61** | 25.27 |
| HAR 11 → 28 | 60.86 | 73.91 | 73.95 | 64.91 | 76.52 | 66.95 | 66.95 | 77.40 | 10.45 | 70.43 | **80.00** | 9.57 | 47.82 | 78.26 | 30.44 | 57.39 | 77.40 | 20.01 |
| HAR 16 → 10 | 50.56 | 40.45 | 71.88 | 68.42 | 56.17 | 61.79 | 35.95 | 68.54 | 32.59 | 62.92 | 72.91 | 9.99 | 61.79 | 78.26 | 16.47 | 67.41 | **87.64** | 20.23 |
| HAR 25 → 10 | 19.10 | 33.70 | 57.81 | 57.89 | 56.18 | 56.18 | 48.31 | 57.40 | 9.09 | 46.06 | 65.17 | 19.11 | 52.80 | **71.91** | 19.11 | 47.19 | 65.17 | 17.98 |
| HAR 18 → 10 | 37.07 | 37.07 | 48.43 | 57.89 | 37.07 | 46.06 | 35.95 | 59.55 | 23.60 | 38.20 | **69.66** | 31.46 | 44.94 | 62.92 | 17.98 | 40.44 | 48.51 | 8.07 |
| HAR 19 → 10 | 44.94 | 52.80 | 50.21 | 49.12 | 39.32 | 43.82 | 37.07 | 49.48 | 12.41 | 42.94 | 46.56 | 3.62 | 66.29 | 64.04 | -2.25 | 37.07 | 50.56 | 13.49 |
| HAR Average | 51.57 | 54.39 | 71.74 | 66.59 | 62.63 | 65.99 | 57.77 | 73.59 | 15.22 | 62.54 | 77.87 | 15.33 | 61.82 | 75.02 | 13.20 | 61.57 | **78.86** | 17.29 |
| WISDM 12 → 9 | 82.71 | 66.67 | 91.35 | 82.50 | 75.30 | 83.95 | 82.50 | 85.19 | 2.69 | 72.83 | **92.60** | 19.77 | 66.67 | 91.36 | 24.69 | 66.67 | 90.12 | 23.45 |
| WISDM 5 → 31 | 59.03 | 68.67 | 80.72 | 82.93 | 75.90 | 82.93 | 75.90 | 74.70 | -1.20 | 81.92 | 81.92 | 0.00 | 67.46 | **93.97** | 26.51 | 65.06 | 83.13 | 18.07 |
| WISDM 25 → 31 | 48.19 | 60.67 | 61.44 | 53.66 | 43.47 | 59.03 | 56.62 | **61.44** | 4.82 | 57.83 | 60.24 | 2.41 | 37.34 | 43.47 | 6.13 | 44.57 | 59.03 | 14.46 |
| WISDM 0 → 30 | 65.04 | 68.93 | 61.16 | 62.75 | 63.10 | 59.03 | 58.22 | 60.19 | 1.97 | 58.22 | 61.16 | 2.94 | 62.13 | **77.66** | 15.53 | 58.25 | 62.14 | 3.89 |
| WISDM 10 → 22 | 61.67 | 73.33 | 73.33 | 76.67 | 51.66 | 71.67 | 71.67 | 76.67 | 5.00 | 73.00 | 76.67 | 3.67 | 66.67 | **91.67** | 25.00 | 56.67 | 78.33 | 21.66 |
| WISDM 12 → 2 | 36.59 | 43.90 | 53.65 | 63.41 | 58.53 | 41.46 | 48.78 | 62.19 | 13.41 | 48.78 | 46.34 | -2.44 | 51.21 | **81.70** | 30.49 | 51.21 | 67.07 | 15.86 |
| WISDM 6 → 11 | 43.42 | 50.00 | 56.57 | 56.10 | 47.36 | 41.46 | 43.36 | 42.10 | -1.26 | 56.57 | 59.21 | 2.64 | 42.10 | 62.10 | 20.00 | 27.63 | **63.15** | 35.52 |
| WISDM 11 → 21 | 28.84 | 59.61 | 38.46 | 58.54 | 40.38 | 30.76 | 18.84 | 19.23 | 0.39 | 38.46 | **59.61** | 21.15 | 30.76 | 58.54 | 27.78 | 17.30 | 55.76 | 38.46 |
| WISDM 19 → 3 | 7.69 | 7.69 | 15.38 | 51.22 | 50.00 | 23.07 | 19.23 | 38.46 | 19.23 | 23.07 | 42.30 | 19.23 | 11.53 | **53.84** | 42.31 | 19.92 | 19.23 | -0.69 |
| WISDM 3 → 11 | 38.16 | 15.78 | 21.36 | 48.78 | 25.00 | 15.78 | 17.10 | 18.42 | 1.32 | 15.78 | 60.52 | 44.74 | 15.79 | 22.36 | 6.57 | 13.15 | 18.42 | 5.27 |
| WISDM Average | 47.13 | 51.52 | 55.34 | 63.66 | 53.79 | 50.91 | 49.22 | 53.86 | 4.64 | 52.65 | 64.06 | 11.41 | 45.17 | **67.67** | 22.50 | 42.04 | 59.64 | 17.60 |
| HHAR 7 → 3 | 55.57 | 54.58 | 94.08 | 85.09 | 79.86 | 86.87 | 61.26 | 88.96 | 27.70 | 83.58 | 94.96 | 11.38 | 74.17 | **95.19** | 21.02 | 84.87 | 92.24 | 7.37 |
| HHAR 6 → 7 | 56.99 | 45.72 | 84.37 | 76.15 | 58.24 | 83.50 | 74.15 | 84.55 | 10.40 | 58.87 | 89.56 | 30.69 | 62.42 | 81.00 | 18.58 | 75.15 | **93.32** | 18.17 |
| HHAR 6 → 3 | 48.14 | 50.10 | 74.33 | 65.79 | 66.52 | 65.86 | 57.55 | 83.58 | 26.03 | 62.36 | 75.27 | 12.91 | 64.55 | 81.83 | 17.28 | 67.36 | **84.47** | 17.11 |
| HHAR 6 → 5 | 45.47 | 61.70 | 75.58 | 45.47 | 61.70 | 45.64 | 47.38 | 66.54 | 19.16 | 49.90 | 70.98 | 21.08 | 49.32 | 75.43 | 26.11 | 42.15 | **76.40** | 34.25 |
| HHAR 7 → 5 | 36.75 | 41.20 | 63.47 | 48.06 | 57.05 | 45.64 | 43.52 | 63.63 | 20.11 | 54.35 | 66.53 | 12.18 | 35.97 | 66.53 | 30.56 | 43.32 | **70.99** | 27.67 |
| HHAR 0 → 7 | 41.97 | 33.89 | 68.32 | 33.89 | 34.34 | 62.83 | 44.25 | **71.81** | 27.56 | 38.20 | 68.47 | 30.27 | 51.15 | 70.35 | 19.20 | 56.57 | 66.97 | 10.40 |
| HHAR 4 → 0 | 22.54 | 23.41 | 23.66 | 34.73 | 25.16 | 25.16 | 23.72 | 24.94 | 1.22 | 23.63 | 28.67 | 5.04 | 21.88 | **37.20** | 15.32 | 27.32 | 31.29 | 3.97 |
| HHAR 3 → 0 | 26.70 | 21.67 | 17.41 | **35.15** | 22.10 | 22.10 | 9.63 | 23.20 | 13.57 | 17.25 | 20.56 | 3.31 | 23.41 | 28.22 | 4.81 | 25.16 | 29.54 | 4.38 |
| HHAR 2 → 7 | 5.42 | 30.27 | 54.68 | 26.36 | 32.98 | 41.33 | 34.65 | 58.89 | 24.24 | 41.54 | **69.10** | 27.56 | 43.63 | 58.03 | 14.40 | 44.25 | 63.63 | 19.38 |
| HHAR 1 → 0 | 36.19 | 40.48 | 57.32 | 47.65 | 45.90 | 44.30 | 41.27 | 58.69 | 17.42 | 44.30 | 60.97 | 16.67 | 19.47 | 31.29 | 11.82 | 48.95 | **64.28** | 15.33 |
| HHAR Average | 37.57 | 40.30 | 61.44 | 49.83 | 48.39 | 52.32 | 43.74 | 62.48 | 18.74 | 47.40 | 64.51 | 17.11 | 44.60 | 62.51 | 17.91 | 51.51 | **67.31** | 15.80 |
| Boiler 1 → 2 | 93.76 | 93.18 | 97.05 | 97.29 | 97.33 | 92.64 | 91.98 | 97.86 | 5.88 | 97.86 | **98.15** | 0.29 | 94.92 | 98.11 | 3.19 | 91.51 | 98.15 | 6.64 |
| Boiler 3 → 2 | 87.16 | 95.65 | 95.02 | 87.16 | 96.05 | 92.17 | 90.83 | **98.15** | 7.32 | 92.17 | 97.84 | 5.67 | 97.68 | 98.11 | 0.43 | 97.47 | 98.15 | 0.68 |
| Boiler Average | 90.46 | 94.41 | 96.03 | 92.22 | 96.69 | 92.41 | 91.41 | 98.00 | 6.59 | 95.02 | 98.00 | 2.98 | 96.30 | 98.11 | 1.81 | 94.49 | **98.15** | 3.66 |

captures temporal properties and reduces correlation shift effectively. Mathematically, the loss function can be expressed as:

$$\mathcal{L} = \mathcal{L}_{\text{c}} + \lambda_{\text{f}}\mathcal{L}_{\text{f}} + \lambda_{\text{c}}\mathcal{L}_{\text{corr}} \tag{12}$$

where $\lambda_{\text{f}}$ and $\lambda_{\text{c}}$ are two hyperparameters

## 5 Experiments

### 5.1 Experimental Settings

**Datasets.** We conduct experiments on 4 real-world datasets, including HAR (Anguita et al., 2013), WISDM (Weiss, 2019), HHAR (Stisen et al., 2015), and Boiler (Cai et al., 2021). For HAR, HHAR, and WISDM datasets, we rank all possible source-target domain pairs based on the magnitude of domain shift, dividing them into 10 groups in the ascending order. From each group, we select one source-target domain pair for evaluation. For Boiler, due to its limited number of domains, we choose the domain pairs with the largest or the smallest domain shift. Detailed descriptions of datasets and domain pair selection are provided in Appendix D and E, respectively.

**Baselines.** We compare CATS with three different types of UDA methods, including (1) correlation-related UDA: CORAL, (2) MTS UDA: SASA, CLUDA, and Raincoat, and (3) adapter-based UDA: UDApter. Descriptions of baseline methods are in Appendix F

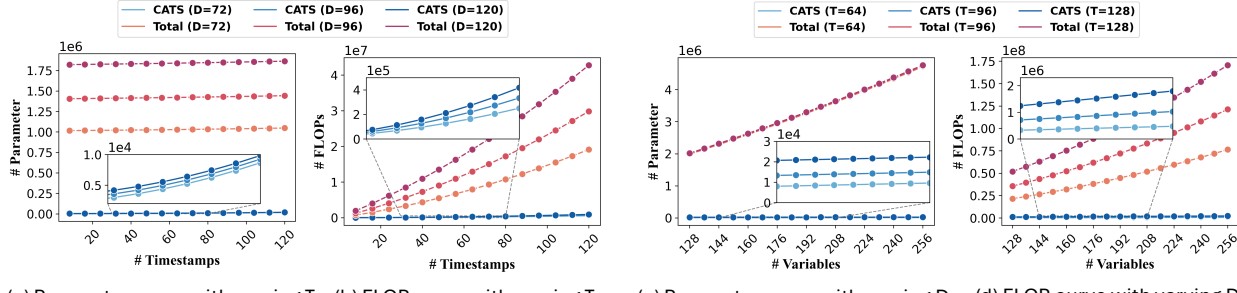

(a) Parameter curve with varying T    (b) FLOP curve with varying T    (c) Parameter curve with varying D    (d) FLOP curve with varying D

Figure 4: The parameter (FLOP) curve of CATS on Transformer with varying variable number $D$ and time length $T$. The three red curves represent the parameter counts (or FLOPs) of the full Transformer model, while the three blue curves represent CATS alone. In Figure (c), the three red curves overlap due to their relatively small differences compared to the large overall values.

**Parameter settings.** Unless otherwise specified, we use default hyperparameter settings in the released code of corresponding publications. For CATS, we use TDCs with a kernel size $r = 5$ and a padding of 2. For training, we use Adam optimizer with a learning rate of 1e-4, and set $\lambda_c = 0.5$ and $\lambda_f = 0.5$. We evaluate the performance of CATS on three different Transformer-based MTS models: Crossformer(Zhang & Yan, 2023), Transformer (Vaswani, 2017), TimesNet (Wu et al., 2022), and iTransformer (Liu et al., 2023). The implementation details are provided in Appendix H.

## 5.2 Experimental Results

**Main results.** The main evaluation results of the accuracy are presented in Table 1. On each dataset in the table, the difficulty of UDA tasks for source-target domain pairs increases progressively from top to bottom. The experimental results reveal three noteworthy conclusions:

- **(1) CATS significantly enhances the UDA classification performance of general MTS models.** Specifically, on four datasets CATS improves the average accuracy of Transformer, TimesNet, Crossformer, and iTransformer on the target domain by 18.56%, 17.60%, 15.80%, and 3.66% accuracy, respectively. Moreover, although CATS are initially designed for Transformers, it still performs well on TimesNet (a CNN model), which highlights CATS's high adaptability across different architectures.
- **(2) CATS shows stable performance even under large shifts.** Even under scenarios with the largest domain shifts, such as HAR 19 → 10 and HHAR 1 → 10, CATS demonstrates robust performance, delivering 6.82% and 15.31% improvement on average for all four models. These results clearly validate the effectiveness of CATS, even in scenarios with large domain shifts.
- **(3) CATS-enhanced MTS models outperform state-of-the-art baselines in classification accuracy.** Across all four datasets, CATS-enhanced models achieve the best performance, with average accuracy improvements of 7.12%, 0.40%, 5.87%, and 1.46% accuracy, respectively, compared to SOTA baselines. These results highlight the superiority of CATS in addressing UDA challenges for MTS data.

**Scalability evaluation.** To validate the scalability of CATS, we adjust the time series length $T$ and the number of variables $D$ of the Transformer, recording the parameter count and FLOPs (Floating Point Operations per Second) for CATS and the full model. The experimental results are shown in Figure 4. The results reveal that, regardless of the values of $T$ and $D$, CATS consistently requires two orders of magnitude fewer parameters and FLOPs compared to the full model. Interestingly, as the hidden layer dimension $D$ increases, the parameter count and FLOPs of the Transformer exhibit quadratic growth, whereas CATS scales linearly. This observation confirms CATS's suitability for large-scale MTS tasks with varying input dimensions.

**Ablation Study.** To validate the effectiveness of CATS and the proposed loss in Eq. (12), we perform ablation studies on the HAR dataset using a Transformer backbone. We first present stepwise results on the

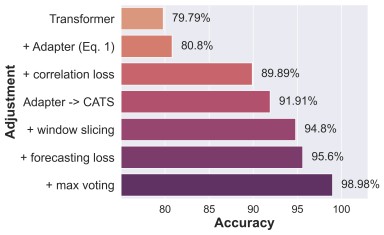

(a) Step-by-Step accuracy improvement on HAR $3 \rightarrow 13$ scenario from vanilla Transformer to Transformer enhanced by CATS.

| Scenario | Vanilla | Fcst | Corr | CATS |
|----------|---------|------|------|------|
| $24 \rightarrow 27$ | 91.81 | 100 | 97.34 | 98.23 |
| $16 \rightarrow 13$ | 73.96 | 76.76 | 77.78 | 77.78 |
| $19 \rightarrow 2$ | 53.84 | 70.33 | 70.33 | 73.52 |
| $16 \rightarrow 10$ | 39.95 | 65.16 | 64.00 | 68.54 |
| $18 \rightarrow 10$ | 35.95 | 57.30 | 58.43 | 59.55 |
| Average | 59.10 | 73.91 | 73.58 | 75.52 |

(b) Loss ablation study. *Vanilla* denotes only using classification loss, *Fcst* introduces additional forecasting loss, *Corr* introduces additional correlation loss, and *CATS* represents using all the losses.

| Scenario | Vanilla | GAT | TDC | CATS |
|----------|---------|-----|-----|------|
| $24 \rightarrow 27$ | 91.81 | 96.46 | 96.46 | 98.23 |
| $16 \rightarrow 13$ | 73.96 | 76.10 | 78.40 | 77.78 |
| $19 \rightarrow 2$ | 53.84 | 70.33 | 72.53 | 73.52 |
| $16 \rightarrow 10$ | 39.95 | 66.29 | 62.91 | 68.54 |
| $18 \rightarrow 10$ | 35.95 | 58.43 | 57.30 | 59.55 |
| Average | 59.10 | 73.52 | 73.52 | 75.53 |

(c) Module ablation study. *Vanilla* represents original models, *GAT / TDC* using a GAT / TDC module as the adapter, and *CATS* denotes using our adapter.

Figure 5: Ablation study on the HAR dataset.

$3 \rightarrow 13$ scenario, incrementally adding components from the vanilla Transformer to the CATS-enhanced version (details in Appendix G). As shown in Figure 5a, each component yields consistent improvements in target domain accuracy. Furthermore, we performed a detailed loss ablation study by fixing the model architecture to the CATS-enhanced Transformer, with results shown in Figure 5b. We also conducted a module ablation study using the full set of loss functions, with results reported in Figure 5c. Together, these analyses comprehensively demonstrate that each design in CATS consistently contributes to improved UDA performance across diverse scenarios.

## 6 Related Works

**Unsupervised Domain Adaptation.** Unsupervised domain adaptation (UDA) leverages labeled data from a source domain to make predictions on an unlabeled target domain and has gained traction across various fields (Ganin & Lempitsky, 2015; Zhang et al., 2018; Ramponi & Plank, 2020; Liu et al., 2021). Existing UDA methods generally fall into three categories: (1) *Adversarial training* uses a domain discriminator to distinguish domains while training the model to extract domain-invariant features (Hoffman et al., 2018; Long et al., 2018; Tzeng et al., 2015); (2) *Multi-task supervision* introduces auxiliary tasks, such as data augmentation (Volpi et al., 2018) or reconstruction (Ghifary et al., 2016; Zhuo et al., 2017), to guide feature learning; (3) *Statistical divergence* methods reduce domain gaps using metrics like MMD (Yan et al., 2017; Zhang & Wu, 2020; Yan et al., 2019), optimal transport (Courty et al., 2017; 2016), or contrastive domain discrepancy (CDD) (Kang et al., 2019). CORAL-based methods (Sun & Saenko, 2016; Lee et al., 2019; Li et al., 2022a) also align feature correlations. However, unlike CATS, these approaches ignore the temporal structure of MTS and the importance of aligning distribution means, as discussed in Appendix I.

**Unsupervised domain adaptation for time series.** While adaptation methods have achieved significant success in computer vision, relatively fewer approaches have been developed to address the unique challenges of domain adaptation for time series data. CoDATS (Wilson et al., 2020) employ domain discriminators for temporal feature alignment. SASA (Cai et al., 2021) aligns invariant unweighted spare associative structures for time series data. RainCoat (He et al., 2023) utilizes MMD to minimize frequency feature distribution in a polar coordinate across domains. Additionally, CLUDA (Ozyurt et al., 2022) leverage contrastive learning to enhance model robustness with data augmentations, while LogoRA (Zhang et al., 2024) combines global and local feature analysis to maintain domain-invariant representations for complex time series structures. More discussion on related works are provided on Appendix L.

## 7 Conclusion

In this paper, we study the problem of unsupervised domain adaptation for multivariate time series classification. We begin by identifying a previously overlooked domain shift in MTS data: correlation shift, where correlations between variables vary across domains. To mitigate this shift, we propose a scalable and parameter-efficient adapter, CATS, serving as a plug-and-play technique compatible with various Transformer variants. Supported by a solid theoretical foundation for mitigating correlation shift, CATS effectively

captures dynamic temporal patterns while adaptively reweighting multivariate correlations. To further reduce correlation discrepancies, we introduce a correlation alignment loss, which aligns multivariant correlations across domains, addressing the non-i.i.d. nature of MTS data. Extensive evaluations on real-world datasets demonstrate that CATS consistently and significantly improves the accuracy of Transformer backbones while introducing minimal additional parameters.

## Broader Impact Statement

This work focuses exclusively on addressing the technical challenge of domain adaptation for multivariate time series classification. All experiments are conducted using publicly available benchmark datasets, ensuring full transparency and reproducibility. Because the study does not involve any human participants, private data, or personally identifiable information, it poses no ethical, privacy, or societal risks.

## GenAI Usage Disclosure

In this manuscript, generative AI tool is used to edit and improve the quality of the text, including checking the spelling, grammar, punctuation and clarity.

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

## A  Empirical Evidence of Correlation Shift

In this section, we conduct an empirical analysis on the Human Activity Recognition (HAR) dataset (Anguita et al., 2013) to demonstrate the prevalence of correlation shift. The HAR dataset consists of 30 domains. Then we iterate through the 30 domains in HAR, treating each domain as the source domain while considering the remaining 29 domains as target domains. Our objective is to examine whether multivariate correlations differ significantly between samples from the source and target domains.

For each sample from either the source or the target domain, we compute the multivariate correlation matrix, which is an $D \times D$ high-dimensional structure with $D$ being the number of variables. However, directly analyzing such high-dimensional matrices is challenging. Hence, we calculate the element-wise mean of the correlation matrices for each domain. Mathematically, if the mean values for the correlation matrices from the source and target domains come from different distributions, it implies a significant difference in the distribution of the overall correlations.

To formally test this, we set the null hypothesis $\mathcal{H}_0$ : *the mean distributions of the source and target domains originate from the same underlying distribution.* We then apply the Mann-Whitney U test (McKnight & Najab, 2010) to verify $\mathcal{H}_0$. If the p-value is less than 0.05, it indicates a statistical confidence of over 95% in rejecting $\mathcal{H}_0$. This rejection implies that the mean values are from different distributions, confirming the statistical significance of the correlation shift between the source and target domains.

Finally, for every source domain, we calculate the rate of target domains with a significant correlation shift among the rest 29 domains. The experiment result is provided in Figure 1. The $x$-axis represents the source domain ID, and the $y$-axis values of the orange bars indicate the rate of target domains with significant correlation shifts. The red dashed line in Figure 1 marks the average rate of correlation shift, which is 78%. These findings provide clear evidence that correlation shifts are prevalent in multivariate time series datasets, highlighting the need to address such shifts.

## B  Empirical Comparison Between TDC and Linear Matrices

To assess the representative learning ability between TDC and linear matrices in adapters, we compare the performance of these two adpaters on HAR dataset with the domain 1 being the source domain and domain 10 being the target domain. Furthermore, we leverege TimesNet as the backbone, and set both the time length $T$ and the hidden dimension $D$ as 128 to make their parameters compatible. Then after pretraining the backbone on the source domain, we only train these adapters on the target domain with accessible labels. We use Adam optimizer with a learning rate of 1e-4 during training.

## C  Theoretical Analysis of Correlation Shift

### C.1  Proof of Correlation Alignment

**Proposition 2** (Correlation Alignment). *Suppose source data $\mathbf{X}_s \in \mathbb{R}^{D \times T}$ and target data $\mathbf{X}_t \in \mathbb{R}^{D \times T}$ follow the source distribution $\mathcal{P}_s(\mathbf{X})$ and the target distribution $\mathcal{P}_t(\mathbf{X})$, respectively. There exists a reweighting matrix $\mathbf{A}^\star \in \mathbb{R}^{D \times D}$, such that the correlation of the source distribution and target distribution can be perfectly aligned, formally expressed as:*

$$\mathrm{Corr}(\mathbf{X}_s) = \mathrm{Corr}(\mathbf{A}^\star \mathbf{X}_t) \tag{6}$$

*Proof.* Let $\boldsymbol{\Sigma}_t = \mathbb{E}_{\mathbf{X}_t \sim \mathcal{P}_t} \left[ (\mathbf{X}_t - \mathbb{E}\mathbf{X}_t)(\mathbf{X}_t - \mathbb{E}\mathbf{X}_t)^\top \right]$ and $\hat{\boldsymbol{\Sigma}}_t = \mathbb{E}_{\mathbf{Y}} \left[ (\mathbf{Y} - \mathbb{E}\mathbf{Y})(\mathbf{Y} - \mathbb{E}\mathbf{Y})^\top \right]$ where $\mathbf{Y} = \mathbf{A}\mathbf{X}_t$. Similarly, let $\boldsymbol{\Sigma}_s = \mathbb{E}_{\mathbf{X}_s \sim \mathcal{P}_s} \left[ (\mathbf{X}_s - \mathbb{E}\mathbf{X}_s)(\mathbf{X}_s - \mathbb{E}\mathbf{X}_s)^\top \right]$. Then based on the spectral theorem, we can decompose the covariance matrices $\boldsymbol{\Sigma}_s$ and $\boldsymbol{\Sigma}_t$:

$$\boldsymbol{\Sigma}_s = \mathbf{U}_s \boldsymbol{\Lambda}_s \mathbf{U}_s^\top \tag{13}$$

$$\boldsymbol{\Sigma}_t = \mathbf{U}_t \boldsymbol{\Lambda}_t \mathbf{U}_t^\top \tag{14}$$

Then since $\mathbf{Y} = \mathbf{A}\mathbf{X}_t$, the covariance matrix $\hat{\boldsymbol{\Sigma}}_t$ could be expressed as

$$\begin{aligned}
\hat{\boldsymbol{\Sigma}}_t &= \mathbf{A}\boldsymbol{\Sigma}_t\mathbf{A}^\top \\
&= \mathbf{A}\mathbf{U}_t\boldsymbol{\Lambda}_t\mathbf{U}_t^\top\mathbf{A}^\top
\end{aligned} \tag{15}$$

Let $\mathbf{A} = \mathbf{U}_s\boldsymbol{\Lambda}_s^{\frac{1}{2}}\boldsymbol{\Lambda}_t^{-\frac{1}{2}}\mathbf{U}_t^\top$. Then we have

$$\begin{aligned}
\hat{\boldsymbol{\Sigma}}_t &= \mathbf{A}\mathbf{U}_t\boldsymbol{\Lambda}_t\mathbf{U}_t^\top\mathbf{A}^\top \\
&= \mathbf{U}_s\boldsymbol{\Lambda}_s^{\frac{1}{2}}\boldsymbol{\Lambda}_t^{-\frac{1}{2}}\mathbf{U}_t^\top\mathbf{U}_t\boldsymbol{\Lambda}_t\mathbf{U}_t^\top\mathbf{U}_t\boldsymbol{\Lambda}_t^{-\frac{1}{2}}\boldsymbol{\Lambda}_s^{\frac{1}{2}}\mathbf{U}_s^\top \\
&= \mathbf{U}_s\boldsymbol{\Lambda}_s\mathbf{U}_s^\top \\
&= \boldsymbol{\Sigma}_s
\end{aligned} \tag{16}$$

Therefore, with the reweighting matrix $\mathbf{A}^\star = \mathbf{U}_s\boldsymbol{\Lambda}_s^{\frac{1}{2}}\boldsymbol{\Lambda}_t^{-\frac{1}{2}}\mathbf{U}_t^\top$, the covariance matrix of $\mathbf{Y}$ on the target domain is equal to the covariance matrix of $\mathbf{X}_s$ on the source domain. Then, since the correlation matrix is only defined by the covariance matrix as shown in Eq. (3), the correlation matrices for $\mathbf{Y}$ on the target domain and $\mathbf{X}_s$ on the source domain are totally the same. □

## C.2 Proof of Gaussian Probability Alignment.

**Proposition 1** (Gaussian Probability Alignment). *Suppose source data $\mathbf{X}_s \in \mathbb{R}^{D \times T}$ and target data $\mathbf{X}_t \in \mathbb{R}^{D \times T}$ follow $\mathcal{N}(\boldsymbol{\mu}_s, \boldsymbol{\Sigma}_s)$ and $\mathcal{N}(\boldsymbol{\mu}_t, \boldsymbol{\Sigma}_t)$, respectively. There exists a reweighting matrix $\mathbf{A}^\star \in \mathbb{R}^{D \times D}$ and a bias vector $\mathbf{b} \in \mathbb{R}^D$, such that the multivariate joint probability of the reweighted target domain perfectly aligns with that of the source domain, that is for every $i, j = 1, 2, ..., D$*

$$\Pr\left(\mathbf{X}_s[i], \mathbf{X}_s[j]\right) = \Pr\left(\mathbf{Y}[i], \mathbf{Y}[j]\right)$$

*where $\mathbf{Y} = \mathbf{A}^\star\mathbf{X}_t + \mathbf{b}$ and $\mathbf{b} = \mathbf{0}$ for most MTS data.*

*Proof.* Based on Proposition 2, it is obvious that $\boldsymbol{\Sigma}_s = \hat{\boldsymbol{\Sigma}}_t$ when $\mathbf{A}^\star = \mathbf{U}_s\boldsymbol{\Lambda}_s^{\frac{1}{2}}\boldsymbol{\Lambda}_t^{-\frac{1}{2}}\mathbf{U}_t^\top$ with $\hat{\boldsymbol{\Sigma}}_t$ being the covariance matrix of $\mathbf{Y}$. Here, $\mathbf{U}_s$ and $\mathbf{U}_t$ are the eigenvector matrices of $\boldsymbol{\Sigma}_s$ and $\boldsymbol{\Sigma}_t$, and $\boldsymbol{\Lambda}_s$ and $\boldsymbol{\Lambda}_t$ are the eigenvalue matrices of $\boldsymbol{\Sigma}_s$ and $\boldsymbol{\Sigma}_t$, respectively. More detailed descriptions are provided in Proposition 2. Then the mean of $\mathbf{Y}$ could be expressed as

$$\mathbb{E}[\mathbf{Y}] = \mathbf{A}\mathbb{E}[\mathbf{X}_t] + \mathbf{b} \tag{17}$$

Therefore, as long as $\mathbf{b} = (\mathbf{I} - \mathbf{A})\mathbb{E}\mathbf{X}_t$, we will have $\mathbb{E}\mathbf{Y} = \mathbb{E}\mathbf{X}_t$. In the real world, it is quite common to normalize the MTS data during data preprocessing (Liu et al., 2023; Wu et al., 2022; Wang et al., 2024b). Under this circumstance, the expectation of normalized data would be zero, and hence $\mathbf{b} = \mathbf{0}$.

Finally, since the covariances and means of two Gaussian distribution is the same, then these two distribution are also the same. Therefore, we have $\Pr(\mathbf{X}_s[i], \mathbf{X}_s[j]) = \Pr(\mathbf{Y}[i], \mathbf{Y}[j])$. □

## C.3 Proof of Attention Approximation

In this section, we prove Theorem 1 (Attention Approximation). To establish the result, we proceed in three steps. First, we introduce two auxiliary lemmas. The first lemma shows that the target function to be approximated is sufficiently smooth, which ensures that it lies in the Barron space. The second lemma demonstrates that a Graph Attention Network (GAT) can naturally degenerate into a one-layer neural network. Finally, by combining these two lemmas with classical Barron approximation results, we derive the desired approximation error bound.

**Lemma 1.** *Let $\mathbf{W} \in \mathbb{R}^{D \times T}$ be a column full-rank matrix. Define*

$$f(\mathbf{w}) = \mathbf{a}^\top\Lambda^{-1/2}(\mathbf{w})U(\mathbf{w})\mathbf{W}\mathbf{b}$$

where $\mathbf{x} \in \mathbb{R}^d$ $(d = D \times T)$ denotes the vectorized form of $\mathbf{W}$, $\Sigma(\mathbf{x}) = (\mathbf{W} - \mathbf{C})(\mathbf{W} - \mathbf{C})^\top$, and $\Sigma(\mathbf{w}) = U(\mathbf{w})\Lambda(\mathbf{w})U(\mathbf{w})^\top$ is the eigendecomposition of $\Sigma(\mathbf{w})$ with eigenvalues sorted in strictly descending order. Here, $\mathbf{a} \in \mathbb{R}^D$ and $\mathbf{b} \in \mathbb{R}^T$ are two constant vectors.

Then for any local smooth branch of the eigenvectors $U(\mathbf{w})$, $f(\mathbf{w})$ is a smooth function and satisfies

$$f \in C^\infty(\mathcal{V})$$

where $\mathcal{V} \subset \mathbb{R}^d$ is the open set consisting of vectorized matrices whose associated $\Sigma(\mathbf{w})$ is positive definite with distinct eigenvalues.

*Proof.* We prove the result by establishing the local smoothness of each mapping involved in the computational graph. Let $\Sigma(\mathbf{w}) = (\mathbf{W} - \mathbf{C})(\mathbf{W} - \mathbf{C})^\top$. We define the open set $\mathcal{V} \subset \mathbb{R}^{DT}$ as the region where $\Sigma(\mathbf{w})$ satisfies

- **Positive definiteness:** all eigenvalues satisfy $\lambda_i > 0$.

- **Simple spectrum:** eigenvalues are strictly distinct. By convention, we globally enforce the ordering $\lambda_1 > \lambda_2 > \cdots > \lambda_T > 0$ on $V$ to uniquely define $\Lambda(\mathbf{w})$.

This region excludes all singularities arising from zero eigenvalues or eigenvalue multiplicities.

**Step 1: Smoothness of the map** $w \mapsto \Sigma(w)$**.** Each entry of $\Sigma$ can be written explicitly as

$$\Sigma_{ij}(\mathbf{w}) = \sum_{k=1}^{D}(w_{ki} - c_{ki})(w_{kj} - c_{kj})$$

Thus every component of $\Sigma$ is a quadratic polynomial in the variables $w_{ij}$. Since multivariate polynomials are infinitely differentiable, the mapping

$$\mathbf{w} \mapsto \Sigma(\mathbf{w})$$

is $C^\infty$ on $\mathbb{R}^{DT}$, and therefore also on the subset $\mathcal{V}$.

**Step 2: Smoothness of eigenvalues** $\Sigma \mapsto \Lambda$**.** Let $\lambda_i$ denote the $i$-th sorted eigenvalue of $\Sigma$. The eigenvalues are the roots of the characteristic equation

$$p(\lambda, \Sigma) = \det(\Sigma - \lambda\mathbf{I}) = 0$$

The function $p(\lambda, \Sigma)$ is a multivariate polynomial in $\lambda$ and the entries of $\Sigma$, and therefore belongs to $C^\infty$. Differentiating with respect to $\lambda$ yields

$$\frac{\partial p}{\partial \lambda}(\lambda_i, \Sigma) = -\prod_{j \neq i}(\lambda_j - \lambda_i)$$

Since eigenvalues are strictly ordered and distinct in $V$, the derivative above is strictly nonzero. Therefore, by the implicit function theorem, each eigenvalue $\lambda_i(\Sigma)$ is a uniquely defined $C^\infty$ function of $\Sigma$ globally on $\mathcal{V}$.

**Step 3: Smoothness of eigenvectors** $\Sigma \mapsto U$**.** Let $\mathbf{u}_i$ denote the unit eigenvector associated with eigenvalue $\lambda_i$. The pair $(\mathbf{u}_i, \lambda_i)$ satisfies the nonlinear system

$$F(\mathbf{u}_i, \lambda_i, \Sigma) = \begin{pmatrix} (\Sigma - \lambda_i\mathbf{I})\mathbf{u}_i \\ \mathbf{u}_i^\top\mathbf{u}_i - 1 \end{pmatrix} = 0$$

The mapping $F$ is polynomial and therefore belongs to $C^\infty$. To apply the implicit function theorem, we compute the Jacobian with respect to $(\mathbf{u}_i, \lambda_i)$:

$$\mathbf{J} = \begin{pmatrix} \Sigma - \lambda_i\mathbf{I} & -\mathbf{u}_i \\ 2\mathbf{u}_i^\top & 0 \end{pmatrix}$$

We now show that $\mathbf{J}$ is invertible. Suppose

$$\mathbf{J}\begin{pmatrix}\mathbf{x}\\ y\end{pmatrix} = 0$$

which yields

$$(\Sigma - \lambda_i \mathbf{I})\mathbf{x} - y\mathbf{u}_i = \mathbf{0}$$

$$2\mathbf{u}_i^\top \mathbf{x} = 0 \implies \mathbf{u}_i^\top \mathbf{x} = 0$$

Left-multiplying the first equation by $\mathbf{u}_i^\top$ gives

$$\mathbf{u}_i^\top(\Sigma - \lambda_i \mathbf{I})\mathbf{x} - y(\mathbf{u}_i^\top \mathbf{u}_i) = 0$$

Since $\Sigma$ is symmetric and $(\Sigma - \lambda_i \mathbf{I})\mathbf{u}_i = 0$, the first term vanishes, yielding

$$-y = 0 \implies y = 0$$

Substituting $y = 0$ back into the first equation gives

$$(\Sigma - \lambda_i \mathbf{I})\mathbf{x} = \mathbf{0}$$

Hence $\mathbf{x}$ lies in the eigenspace of $\lambda_i$. Because $\lambda_i$ is simple on $\mathcal{V}$, this eigenspace is one-dimensional, implying $\mathbf{x} = c\mathbf{u}_i$. Substituting this into $\mathbf{u}_i^\top \mathbf{x} = 0$, we obtain $c(\mathbf{u}_i^\top \mathbf{u}_i) = c = 0$, and thus $\mathbf{x} = \mathbf{0}$. Therefore the null space of $\mathbf{J}$ is trivial, implying $\mathbf{J}$ is invertible. Because $\mathbf{u}_i$ is only determined up to a sign $\pm 1$, $U(\mathbf{w})$ is not globally unique. However, by the implicit function theorem, for any point in $\mathcal{V}$ and a valid initial sign choice, there exists a neighborhood where the eigenvector $\mathbf{u}_i(\Sigma)$ forms a smooth branch. Consequently, the orthogonal matrix $U(\Sigma)$ is locally $C^\infty$.

**Step 4: Smoothness of $\Lambda^{-1/2}$ and final composition.** The mapping $\Lambda \mapsto \Lambda^{-1/2}$ applies the scalar function

$$g(x) = x^{-1/2}$$

to each eigenvalue. Since all eigenvalues satisfy $\lambda_i > 0$ on $V$, the function $g(x)$ is infinitely differentiable on $(0, \infty)$, implying that the matrix mapping $\Lambda \mapsto \Lambda^{-1/2}$ is also $C^\infty$. Finally, given a local smooth branch of $U(\mathbf{w})$, the function

$$f(w) = \mathbf{a}^\top \Lambda^{-1/2}(\mathbf{w})U(\mathbf{w})\mathbf{W}\mathbf{b}$$

consists of matrix multiplications and inner products, which are smooth operations.

**Conclusion.** The entire computational pipeline for a locally chosen smooth branch of eigenvectors can be summarized as

$$\mathbf{w} \xrightarrow{C^\infty} \Sigma \xrightarrow{C^\infty} (\Lambda, U) \xrightarrow{C^\infty} (\Lambda^{-1/2}, U) \xrightarrow{C^\infty} f(\mathbf{w})$$

Since the composition of $C^\infty$ functions remains $C^\infty$, we conclude that for any local smooth choice of $U(\mathbf{w})$,

$$f \in C^\infty(\mathcal{V})$$

$\square$

**Lemma 2.** *Given the input $\mathbf{X} \in \mathbb{R}^{D \times T}$, a Graph Attention Network (GAT) can naturally degenerate into a one-layer neural network: $f(\mathbf{x}) = \sum_{i=1}^{D} w_i \sigma(\mathbf{a}_i^\top \mathbf{x})$, where $\mathbf{x}$ is the flattened vector of the matrix $\mathbf{X}$. Formally, the GAT follows:*

$$\mathrm{GAT}(\mathbf{X}) = \mathbf{A}\mathbf{X}\mathbf{W}, \quad \mathbf{A}[i,j] = \sigma\big(\mathbf{a}^\top [\mathbf{W}^T \mathbf{X}[i] \,\|\, \mathbf{W}^T \mathbf{X}[j]]\big)$$

*where $\sigma$ is the non-linear activation function, $\mathbf{W} \in \mathbb{R}^{T \times T}$ is a learnable linear matrix.*

*Proof.* First, we define the concatenated input vector $\mathbf{x} := \|_{k=1}^{D} \mathbf{X}[k] \in \mathbb{R}^{DT}$, which flattens the input matrix $\mathbf{X}$ into a single vector. Next, we construct the inner weight vectors to compute the attention coefficients. For any specific pair of nodes $(i, j)$, we define a parameter vector $\mathbf{a}'_{i,j} \in \mathbb{R}^{DT}$ as:

$$\mathbf{a}'_{i,j} := \|_{k=1}^{D} (\mathbf{W}\mathbf{a}_1 \mathbb{1}(k=i) + \mathbf{W}\mathbf{a}_2 \mathbb{1}(k=j))$$

where $\mathbf{a} = [\mathbf{a}_1 \| \mathbf{a}_2]$ splits the attention mechanism's weight vector. With this construction, the inner product perfectly extracts and projects the $i$-th and $j$-th node features:

$$(\mathbf{a}'_{i,j})^\top \mathbf{x} = \mathbf{a}_1^\top \mathbf{W}^\top \mathbf{X}[i] + \mathbf{a}_2^\top \mathbf{W}^\top \mathbf{X}[j]$$

Thus, the attention coefficient can be rewritten as a standard neuron activation: $\mathbf{A}[i,j] = \sigma\left((\mathbf{a}'_{i,j})^\top \mathbf{x}\right)$. Now, let $q_{i,j}(\mathbf{x})$ denote the function computing the $(i,j)$-th element in the output matrix GAT($\mathbf{X}$). By expanding the matrix multiplication, we get:

$$q_{i,j}(\mathbf{x}) = \sum_{k=1}^{D} \mathbf{A}[i,k] \cdot (\mathbf{XW})[k,j]$$

Let us define the outer weights as $w_{k,j} := (\mathbf{XW})[k,j]$, and $w_{k,j}$ is an linear transformation of the input vector $\mathbf{x}$. Substituting $\mathbf{A}[i,k]$, we obtain:

$$q_{i,j}(\mathbf{x}) = \sum_{k=1}^{D} w_{k,j} \cdot \sigma\left((\mathbf{a}'_{i,k})^\top \mathbf{x}\right)$$

This equation structurally matches a one-layer neural network. However, unlike a standard shallow network whose outer weights are constant parameters, the outer weights here $w_{k,j} := (\mathbf{XW})[k,j] = (\mathbf{W}[j])^T \mathbf{X}[k]$ depend explicitly on the input, and thus the resulting GAT follows the more general form of $\sum (\mathbf{b}^\top \mathbf{x}) \sigma(\mathbf{a}^\top \mathbf{x})$. Consequently, the degenerated GAT corresponds to a *GLU-style multiplicative (gated) network*, and each neuron of such the network can naturally represents "conditional modulation" via two input-dependent features. Recent advances in neural approximation theory (Li & Lu, 2024; Ma et al., 2022; Ben-Shaul et al., 2023; Siegel & Xu, 2020) show that such GLU-style multiplicative network can approximate the functions from the extended Barron space beyond the classical Barron space, and hence exhibits stronger expressiveness than standard one-layer neural networks (Shazeer, 2020; Dauphin et al., 2017). By incorporating multiplicative interactions between affine input terms and nonlinear activations, these models possess stronger expressive power than ordinary Barron functions.

To illustrate the advantage, consider a quadratic target function $f(\mathbf{x}) = \mathbf{x}^\top \mathbf{Q} \mathbf{x}$. Approximating such second-order interactions using a standard one-layer network generally requires a large number of neurons due to its purely additive structure. In contrast, under our formulation, the linear components of $\mathbf{x}$ are explicitly extracted into the outer weights $w_{k,j}$, allowing the multiplicative interaction with the activation function. As a result, the network can approximate quadratic or higher-order feature interactions—commonly observed in multivariable data—with significantly fewer neurons and a smaller approximation constant. $\square$

**Theorem 1** (Attention Approximation). *Let $\mathbf{A}^\star$ denote the optimal reweighting matrix defined in Proposition 2. Suppose the input matrix $\mathbf{X} \in \mathbb{R}^{D \times T}$ is drawn from a high-dimensional distribution with probability measure $P$. Assume that the following conditions hold:(1) $\mathbf{X}$ has full column rank;(2) The covariance matrix of $\mathbf{X}$ exists and possesses strictly distinct eigenvalues. Under these conditions, there exists a one-layer Graph Attention Network with a hidden dimension of $D$ such that*

$$\mathbb{E}_P \left[ \text{MSE} \left( \text{GAT} \left( \mathbf{X} \right) - \mathbf{A}^\star \mathbf{X} \right) \right] \leq \frac{C}{D} \tag{7}$$

*where $C > 0$ are a constant, and GAT follows the following formula:*

$$\text{GAT}(\mathbf{X}) = \mathbf{AXW}, \quad \mathbf{A}[i,j] = \sigma\left(\mathbf{a}^\top \left[\mathbf{W}^T \mathbf{X}[i] \| \mathbf{W}^T \mathbf{X}[j]\right]\right) \tag{8}$$

*where $\sigma : \mathbb{R} \to \mathbb{R}$ is a non-linear activation function, $\mathbf{A}[i,j]$ denote the $(i,j)$-th entry of $\mathbf{A}$, $\mathbf{W} \in \mathbb{R}^{T \times T}$ is a learnable linear projection. $\mathbf{a} \in \mathbb{R}^{2T}$ is a learnable attention vector, and $\|$ denotes vector concatenation. Consequently, as the input dimension $D$ approaches infinity, the GAT output converges to the optimal linear transformation $\mathbf{A}^\star \mathbf{X}$ in the mean squared sense:*

$$\lim_{D \to \infty} \mathbb{E}_P \left[ \text{MSE} \left( \text{GAT}(\mathbf{X}) - \mathbf{A}^\star \mathbf{X} \right) \right] = 0.$$

*Proof.* We consider matrices $\mathbf{X} \in \mathbb{R}^{D \times T}$. For convenience, we vectorize $\mathbf{X}$ into $\mathbf{x} \in \mathbb{R}^d$ with $d = D \cdot T$. The input $\mathbf{x}$ is assumed to follow a high-dimensional distribution with probability measure $P$.

**Step 1. Smoothness of the target function.** First, we set the target function as the mapping of learning the $(i,j)$-th element of $\mathbf{A}^\star \mathbf{X}$, and will show that (the surrogate of) this function is smooth enough to fall into the Barron space that could be easily approximated by a one-layer neural network.

Define the matrix $\Sigma(\mathbf{x}) = (\mathbf{X} - \mathbf{C})^\top (\mathbf{X} - \mathbf{C})$, and its spectral decomposition $\Sigma(\mathbf{x}) = U(\mathbf{x})\Lambda(\mathbf{x})U(\mathbf{x})^\top$, where $\Lambda(\mathbf{x}) = \mathrm{diag}(\lambda_1, \dots, \lambda_D)$ contains the eigenvalues and $U(w)$ contains the corresponding orthonormal eigenvectors. Then, the target function in Proposition 2 is defined as

$$f(\mathbf{x}) = \mathbf{a}^\top \Lambda^{-1/2}(\mathbf{x}) U(\mathbf{x}) \mathbf{X} \mathbf{b}$$

where $\mathbf{a}$ and $\mathbf{b}$ are two vectors. Clearly, this function $f(\mathbf{x})$ becomes ill-defined when the eigendecomposition loses smoothness. We define the singular set $\mathcal{S} \subset \mathbb{R}^d$ as:

$$\mathcal{S} = \{\mathbf{x} : \lambda_{\min}(\Sigma(\mathbf{x})) \le 0\} \cup \{\mathbf{x} : \exists i \ne j, \ \lambda_i(\Sigma(\mathbf{x})) = \lambda_j(\Sigma(\mathbf{x}))\} \tag{18}$$

Furthermore, we restrict the domain away from $\mathcal{S}$ and truncate the function outside a compact set.

**Definition 2.** *Let $R > 0$ and $\epsilon_1, \epsilon_2 > 0$. Define the safe region*

$$\Omega = \left\{ \mathbf{x} \in \mathbb{R}^d \ \middle| \ \|\mathbf{x}\|_2 \le R, \ \lambda_{\min}(\Sigma(\mathbf{x})) \ge \epsilon_1, \ \min_{i \ne j} |\lambda_i - \lambda_j| \ge \epsilon_2 \right\}.$$

The set $\Omega$ excludes both singular matrices and eigenvalue degeneracies while keeping the domain bounded. Furthermore, based on Urysohn's lemma, there exists an open set $\mathcal{V}$ such that $\Omega \subset \mathcal{V}$, $V \cap \mathcal{S} = \varnothing$, and a smooth compactly-supported function $\Psi : \mathbb{R}^d \to [0,1]$ satisfying

$$\Psi(\mathbf{x}) = 1 \quad \forall w \in \Omega,$$

$$\Psi(\mathbf{x}) = 0 \quad \forall w \notin \mathcal{V},$$

with $\Psi \in C_c^\infty(\mathbb{R}^d)$. Using the cutoff function $\Psi(\mathbf{x})$, we define a surrogate function that coincides with $f$ on $\Omega$ but remains globally smooth.

**Definition 3** (Smooth Surrogate Function).

$$\hat{f}(\mathbf{x}) = f(\mathbf{x})\Psi(\mathbf{x}) = \left( \mathbf{a}^\top \Lambda^{-1/2}(\mathbf{x}) U(\mathbf{x}) \mathbf{b} \right) \Psi(\mathbf{x}).$$

*For $\mathbf{x} \notin V$ we define $\hat{f}(\mathbf{x}) = 0$.*

Since $f(\mathbf{x}) \in C^\infty(\mathcal{V})$ based on Lemma 1, $\hat{f} \in C_c^\infty(\mathbb{R}^d)$.

Then, we aim to show that the surrogate function $\hat{f}$ belongs to the Barron space so that it could be easily approximated by a one-layer neural network. By the Paley–Wiener theorem, the Fourier transform $\tilde{\hat{f}}(\omega)$ of a compactly supported smooth function exhibits super-polynomial decay. Specifically, for any integer $k > 0$ there exists a constant $C_k$ such that

$$|\tilde{\hat{f}}(\omega)| \le \frac{C_k}{(1 + \|\omega\|_2)^k}.$$

The Barron norm is defined as $C_{\hat{f}} = \int_{\mathbb{R}^d} \|\omega\|_2 |\tilde{\hat{f}}(\omega)| \, d\omega$. Choosing $k = d + 2$, the integrand behaves as $O(\|\omega\|^{-(d+1)})$, which is integrable in $\mathbb{R}^d$. Therefore, $C_{\hat{f}} < \infty$, and $\hat{f}$ lies in the Barron space.

**Step 2. Error Computation.** Since $\hat{f}$ belongs to the Barron space, the Barron approximation theorem implies the existence of a one-layer neural network $f_{NN}$ with $n$ neurons such that

$$\mathbb{E}_{\mathbf{x} \sim P}\left[(f_{NN}(\mathbf{x}) - \hat{f}(\mathbf{x}))^2\right] \leq \frac{(2C_{\hat{f}})^2}{n}.$$

Furthermore, according to Lemma 2, such a one-layer neural network can be equivalently implemented by a single-layer Graph Attention Network. In this construction, the number of hidden neurons corresponds to the number of attention channels $D$, i.e., $n = D$.

We are interested in approximating the original function $f$. Define the expected risk as:

$$\mathcal{R} = \mathbb{E}_{\mathbf{x} \sim P}\left[(f_{NN}(\mathbf{x}) - f(\mathbf{x}))^2\right].$$

Since $\mathbb{E}_P[(f_{NN} - f)^2] \leq 2\mathbb{E}_P[(f_{NN} - \hat{f})^2] + 2\mathbb{E}_P[(\hat{f} - f)^2]$, The expected risk satisfies

$$\mathcal{R} \leq \frac{8C_{\hat{f}}^2}{D} + 2\int_{\Omega^c}(f_{NN}(\mathbf{x}) - f(\mathbf{x}))^2\, dP(\mathbf{x}).$$

The remaining error term arises from the probability mass outside the safe region $\Omega$. Applying the Cauchy–Schwarz inequality,

$$\int_{\Omega^c}(f_{NN} - f)^2 \cdot 1 dP \leq \left(\int_{\Omega^c}(f_{NN} - f)^4 dP\right)^{1/2}\left(\int_{\Omega^c}1^2 dP\right)^{1/2}$$

Let $\int_{\Omega^c}1dP = P(\Omega^c)$ and $M = \left(\mathbb{E}_P[(f_{NN} - f)^4]\right)^{1/2}$. Therefore the final risk bound becomes

$$\mathbb{E}_{\mathbf{x} \sim P}[(f_{NN}(\mathbf{x}) - f(\mathbf{x}))^2] \leq \frac{8C_{\hat{f}}^2}{D} + 2M\sqrt{P(\Omega^c)}.$$

In the previous analysis, the tail probability $P(\Omega^c)$ plays a key role in the overall risk bound. Since the covariance of the matrix $\mathbf{X}$ exists, the distribution $P$ of the matrix $\mathbf{X}$ satisfies the finite 2-th moment condition:

$$\mathbb{E}_P\left[\|\mathbf{X}\|_F^2\right] = M_2 < \infty.$$

Using Markov's inequality, the probability that the Frobenius norm of $W$ exceeds a threshold $R$ can be bounded as

$$P(\Omega^c) \leq \frac{\mathbb{E}[\|\mathbf{X}\|_F^2]}{R^2} = \frac{M_2}{R^2}.$$

Therefore, we have $\mathbb{E}_{\mathbf{x} \sim P}[(f_{NN}(\mathbf{x}) - f(\mathbf{x}))^2] \leq \frac{C_1}{D} + \frac{C_2}{R}$. Given any $\epsilon > 0$, let $R \geq \frac{C_2}{\epsilon}$, then $\mathbb{E}_{\mathbf{x} \sim P}[(f_{NN}(\mathbf{x}) - f(\mathbf{x}))^2] \leq \frac{C_1}{D} + \epsilon$.

**Step 3. Matrix-form Transformation.** Let the target error matrix $E = \text{GAT}(\mathbf{X}) - \mathbf{A}^{\star}\mathbf{X} \in \mathbb{R}^{D \times T'}$. Then for any $(i, j)$-th element of $E$, we have $\mathbb{E}_P[E_{i,j}^2] \leq \frac{C_{i,j}^2}{D}$. Therefore, the matrix-form error bound can be expressed as

$$\mathbb{E}_P[\text{MSE}(E)] = \frac{1}{D \cdot T'}\sum_{i=1}^{D}\sum_{j=1}^{T'}\mathbb{E}_P[E_{i,j}^2] \leq \frac{1}{D \cdot T'}\sum_{i=1}^{D}\sum_{j=1}^{T'}\frac{C_{i,j}^2}{D} \leq \frac{C}{D}$$

where $C = \max_{i,j} C_{i,j}^2$. This completes the proof.

$\square$

### C.4 Proof of CATS Approximation

**Theorem 2** (CATS Approximation). *Given a linear CATS module in Eq. (9) and a input matrix $\mathbf{X}$ drawn from a high-dimensional distribution with probability measure $P$, i.e.,*

$$\phi(\mathbf{X}) = \mathbf{X} + \mathrm{TDC}_\uparrow \left( \mathrm{GAT} \left( \mathrm{TDC}_\downarrow \left( \mathbf{X} \right) \right) \right), \tag{19}$$

*then $\phi(\mathbf{X})$ could approximate the optimal transformation $\mathbf{Y} = \mathbf{A}^\star \mathbf{X}$ in Propositions 2, when (1) $\mathbf{X}$ has full column rank, and (2) the covariance matrix of $\mathbf{X}$ exists and possesses strictly distinct eigenvalues. Specifically, there exist constants $C > 0$ such that*

$$\mathbb{E}_p \left[ \mathrm{MSE} \left( \phi(\mathbf{X}) - \mathbf{A}^\star \mathbf{X} \right) \right] \leq \frac{C}{D} \tag{20}$$

*where $D$ is the input dimension of the CATS module.*

*Proof.* When using a depthwise convolution with a stride of 1, zero-padding, and a convolution kernel where only the first element is 1 while all others are 0, the convolution operation automatically degenerates into the identity mapping $f(x) = x$. Under this circumstance, Eq. (19) would be further simplified as

$$\phi(\mathbf{X}) = \mathbf{X} + \mathrm{GAT}(\mathbf{X}) \tag{21}$$

Then, since Theorem 1 has no requirement on the approximated matrix $\mathbf{A}^\star$, we could leverage a GAT layer with a hidden dimension of $m$ to approximate the matrix $\mathbf{A} - \mathbf{I}$:

$$\mathbb{E}_p \left[ \mathrm{MSE} \left( \mathrm{GAT}(\mathbf{X}) - (\mathbf{A}^\star - \mathbf{I})\mathbf{X} \right) \right] \leq \frac{C}{D} \tag{22}$$

Therefore, the formula of CATS would be expressed as

$$\mathbb{E}_p \left[ \mathrm{MSE} \left( \phi(\mathbf{X}) - \mathbf{A}^\star \mathbf{X} \right) \right] \leq \frac{C}{D} \tag{23}$$

$\square$

## D   Dataset Description

Table 2: The statistics of datasets.

| Dataset | # Domains | # Timestamps | # Variables |
|---------|-----------|--------------|-------------|
| HAR | 30 | 128 | 9 |
| WIDSM | 36 | 128 | 3 |
| HHAR | 9 | 128 | 3 |
| Boiler | 3 | 128 | 20 |

In this paper, we validate the effectiveness of CATS on four different datasets, HAR (Anguita et al., 2013), WISDM (Weiss, 2019), HHAR (Stisen et al., 2015), and Boiler (Cai et al., 2021). The statistics of datasets are provided in Table 2, and the detailed information is listed below.

- **HAR dataset**. The Human Activity Recognition Dataset has been collected from 30 subjects performing six different activities (Walking, Walking Upstairs, Walking Downstairs, Sitting, Standing, Laying). It consists of inertial sensor data that was collected using a smartphone carried by the subjects.

- **WISDM dataset**. WISDM Smartphone and Smartwatch Activity and Biometrics Dataset collects raw accelerometer and gyroscope sensor data from the smartphone and smartwatch at a rate of 20Hz. It is collected from 51 test subjects as they perform 18 activities for 3 minutes apiece.

Table 3: The Wasserstein distance between selected domain pairs from four real-world datasets.

| dataset | HAR | | | | | | | | | | WISDM | | | | | | | | | |
|---|---|---|---|---|---|---|---|---|---|---|---|---|---|---|---|---|---|---|---|---|
| Source → Target | 24→27 | 3→13 | 16→13 | 3→8 | 19→2 | 11→28 | 16→10 | 25→10 | 18→10 | 19→10 | 12→9 | 5→31 | 25→31 | 0→30 | 10→22 | 12→2 | 6→11 | 11→21 | 19→3 | 3→11 |
| Wass. Distance | 0.26 | 0.31 | 0.35 | 0.47 | 0.48 | 0.48 | 0.52 | 0.56 | 0.59 | 0.63 | 1.35 | 1.38 | 1.40 | 1.44 | 1.43 | 2.24 | 2.29 | 2.35 | 2.55 | 2.55 |

| dataset | HHAR | | | | | | | | | | FD | |
|---|---|---|---|---|---|---|---|---|---|---|---|---|
| Source → Target | 7→3 | 6→7 | 6→3 | 6→5 | 7→5 | 0→7 | 4→0 | 3→0 | 2→7 | 1→0 | 1→2 | 3→2 |
| Wass. Distance | 1.34 | 1.67 | 1.80 | 2.01 | 2.27 | 2.45 | 2.54 | 2.71 | 2.93 | 3.22 | 0.60 | 0.71 |

- **HHAR dataset**. The Heterogeneity Dataset for Human Activity Recognition contains the readings of two motion sensors commonly found in smartphones. Reading were recorded while nine users executed six different activities scripted in no specific order carrying smartwatches and smartphones.

- **Boiler dataset**. The boiler data consists of sensor data from three boilers from 2014/3/24 to 2016/11/30. There are 3 boilers in this dataset and each boiler is considered as one domain. We slice the original time series data with a time window of 128 and a stride of 32.

## E  Domain Pair Selection

In this study, we utilize four datasets, each containing a large number of domains. As a result, exhaustively evaluating all possible source-target domain pairs is impractical (for example, 900 pairs for the HAR dataset). Therefore, selecting reasonable and effective source-target domain pairs becomes critically important.

To address this, we adopt the following domain pair selection mechanism: For each source-target domain pair, we compute the Wasserstein distance between samples sharing the same label in the source and target domains. We then sum the distances across all possible labels. Mathematically, this distance can be expressed as:

$$d = \sum_{y \in \mathcal{Y}} \text{Wass}(\mathcal{P}_S^y, \mathcal{P}_T^y) \tag{24}$$

where $\mathcal{P}_S^y$ and $\mathcal{P}_T^y$ represent the distributions of samples with label $y$ in the source domain $S$ and target domain $T$, respectively, and $\text{Wass}(\cdot, \cdot)$ denotes the Wasserstein distance. This distance $d$ quantifies the similarity between the source and target domains: the smaller the distance, the smaller the domain shift, and the lower the difficulty of domain adaptation.

For HAR, HHAR and WISDM datasets, we divide all domain pairs into 10 groups, sorted by increasing the distance $d$. From each group, we sample one domain pair. This strategy ensures that the selected domain pairs represent varying levels of domain adaptation difficulty, from small to large domain shifts. For the Boiler dataset, due to its quite limited domain pairs (3 domains and 6 domain pairs in total), we only choose the domain pair with the largest $d$ and the smallest $d$, respectively. The detailed Wasserstein distances between those selected domain pairs are provided in Table 3.

The experimental results, summarized in Table 1, demonstrate the performance of our method across these selected domain pairs. Note that within each dataset, the domain pairs from the top to the bottom in Table 1 are ordered by increasing $d$, indicating progressively higher domain adaptation difficulty (e.g., in the HAR dataset, the pair $24 \rightarrow 27$ represents the smallest difficulty, while $19 \rightarrow 10$ represents the largest difficulty).

## F  Description of Baselines

In this paper, we compare CATS with 5 different baselines. These baselines could be roughly divided into three different categories.

First, correlation-related UDA method is

- **CORAL** (Sun & Saenko, 2016) learn a nonlinear transformation that aligns correlations of layer activations in deep neural networks.

Second, MTS-related UDA methods include

- **Raincoat** (He et al., 2023) uses time and frequency-based encoders on the polar coordinate of frequency to learn domain-invariant time series representations.

- **SASA** (Cai et al., 2021) introduces the intra-variables and inter-variables sparse attention mechanisms to extract associative structure time-series data with considering time lags for domain adaptation.

- **CLUDA** (Ozyurt et al., 2022) proposes a contrastive learning framework to learn domain-invariant, contextual representation for UDA of time series data.

Third, we introduce an adapter-related UDA method:

- **UDApter** (Malik et al., 2023) adds a domain adapter to learn domain-invariant information and a task adapter that uses domain-invariant information to learn task representations in the source domain.

## G Step-by-step Incremental Adjustment

In the ablation study, we progressively adjusted the vanilla Transformer to the CATS-enhanced Transformer, resulting in a significant improvement in accuracy from 79.79% to 98.98%. Specifically, we introduced the following six incremental adjustments:

1. **+ Adapter (Eq. 1).** We incorporate the adapter defined in Eq. (1) into the vanilla Transformer and trained it using the classification loss function $\mathcal{L}_c$ in Eq. 11 on the source domain. This modification results in an accuracy improvement of 1.01%.

2. **+ correlation loss.** We optimize the adapter using a combination of classification loss and correlation alignment loss. This step further enhances accuracy by 9.89%.

3. **Adapter $\rightarrow$ CATS.** We replace the adapter in Eq. 1 with CATS and train it with the combined classification and correlation alignment loss. This substitution improved accuracy by 2.02%.

4. **+ window slicing.** To align with the setting of forecasting loss, we slice the original samples with a length of 128 into overlapping time windows with a length of 48 and used these sliced windows as inputs to train CATS. This adjustment yields an additional accuracy gain of 2.89%.

5. **forecasting loss.** We introduce the forecasting loss, which uses consecutive time windows as input and their corresponding ground truth for prediction. The final loss function $\mathcal{L}$ in Eq. (**??**) is then leveraged to train CATS, resulting in an accuracy improvement of 0.8%.

6. **+ max voting.** We apply a max-voting method to assign the label of the original sample based on predictions from its sliced time windows. This final step further boosted accuracy by 3.32%.

## H Implementation Details

Table 4: Hyperparameters of backbone models.

| Hyperparameter | e_layers | d_model | d_ff | top_k | epoch (pretrain) |
|---|---|---|---|---|---|
| Value | 3 | 128 | 256 | 3 | 10 |

We use the code from Time-Series-Library repository [4] to construct three different Transformer variants as backbone models, Transformer, TimesNet, and iTransformer. The hyperparameters for these three models

---

[4]`https://github.com/thuml/Time-Series-Library`

follow the default configuration on Time-Series-Library repository, as shown in Table 4. For CATS, we use the TCNs with a kernel size $r = 5$ and a padding of 2. We use Xavier initialization for the down-project TDC and GAT, and zero initialization for the up-down TDC. For training, we set the length of sliced time windows as 48 and set the number of sampled windows $m$ for max voting as 16. We use Adam optimizer with a learning rate of 1e-4, and set $\lambda_{\mathtt{c}} = 0.5$ and $\lambda_{\mathtt{f}} = 0.5$.

## I   Comparison Between Correlation Alignment Loss and CORAL Loss

CORAL loss (Sun & Saenko, 2016) is one widely-used domain adaptation loss, which focuses on minimizing the covariance between the source samples and the target samples. In this section, we will demonstrate that the correlation alignment loss offers advantages over the CORAL loss. Furthermore, we show that under certain simplified conditions, the correlation alignment loss can be reduced to the CORAL loss, providing a unified perspective on both approaches.

Our correlation alignment loss aim to use MMD to minimize the mean of the distributions of $\mathrm{corr}\left(\mathbf{H}^{s}\right)$ and $\mathrm{corr}\left(\mathbf{H}^{\top}\right)$. Let the distributions of $\mathrm{corr}\left(\mathbf{H}^{s}\right)$ and $\mathrm{corr}\left(\mathbf{H}^{\top}\right)$ be denoted as $\mathcal{C}^{s}$ and $\mathcal{C}^{\top}$, respectively. Mathematically, we aim to optimize the following equation.

$$
\begin{aligned}
\mathcal{L}_{\mathtt{corr}} &= \mathrm{MMD}(\mathcal{C}^{s}, \mathcal{C}^{\top}) \\
&= \left\| \mathbb{E}\left[\psi\left(\mathrm{corr}\left(\mathbf{H}^{s}\right)\right)\right] - \mathbb{E}\left[\psi\left(\mathrm{corr}\left(\mathbf{H}^{\top}\right)\right)\right] \right\|_{2}
\end{aligned}
\tag{25}
$$

where $\psi(\cdot)$ is one feature mapping function and $\mathrm{corr}(\mathbf{H}) = \mathrm{vec}\left(\frac{(\mathbf{H})(\mathbf{H})^{\top}}{\|\mathbf{H}\|_{\mathcal{F}}^{2}}\right)$. Here, let us relax this feature mapping function to be the identity function, i.e., $\psi(\mathbf{X}) = \mathbf{X}$. Then our optimization objective could be further deduced:

$$
\begin{aligned}
\mathcal{L}_{\mathtt{corr}} &= \left\| \mathbb{E}\left[\mathrm{corr}\left(\mathbf{H}^{s}\right)\right] - \mathbb{E}\left[\mathrm{corr}\left(\mathbf{H}^{\top}\right)\right] \right\|_{2} \\
&= \left\| \mathbb{E}\left[\hat{\mathbf{h}}^{s}(\hat{\mathbf{h}}^{s})^{\top}\right] - \mathbb{E}\left[\hat{\mathbf{h}}^{\top}(\hat{\mathbf{h}}^{\top})^{\top}\right] \right\|_{2} \\
&= \left\| \mathbb{E}\left[\left(\hat{\mathbf{h}}^{s} - \mathbb{E}\left[\hat{\mathbf{h}}^{s}\right]\right)\left(\hat{\mathbf{h}}^{s} - \mathbb{E}\left[\hat{\mathbf{h}}^{s}\right]\right)^{\top}\right] \right. \\
&\quad - \mathbb{E}\left[\left(\hat{\mathbf{h}}^{\top} - \mathbb{E}\left[\hat{\mathbf{h}}^{\top}\right]\right)\left(\hat{\mathbf{h}}^{\top} - \mathbb{E}\left[\hat{\mathbf{h}}^{\top}\right]\right)^{\top}\right] \\
&\quad \left. + \mathbb{E}\left[\hat{\mathbf{h}}^{s}\left(\hat{\mathbf{h}}^{s}\right)^{\top}\right] - \mathbb{E}\left[\hat{\mathbf{h}}^{\top}\left(\hat{\mathbf{h}}^{\top}\right)^{\top}\right] \right\|_{2}
\end{aligned}
\tag{26}
$$

where $\hat{\mathbf{h}}^{s}$ and $\hat{\mathbf{h}}^{\top}$ are the normalized vector from $\mathrm{vec}(\mathbf{H}^{s})$ and $\mathrm{vec}(\mathbf{H}^{\top})$, i.e., $\hat{\mathbf{h}}^{s} = \frac{\mathrm{vec}(\mathbf{H}^{s})}{\|\,\mathrm{vec}(\mathbf{H}^{s})\|_{2}}$ and $\hat{\mathbf{h}}^{\top} = \frac{\mathrm{vec}(\mathbf{H}^{\top})}{\|\,\mathrm{vec}(\mathbf{H}^{\top})\|_{2}}$. Due to the triangle inequality, we have

$$
\begin{aligned}
\mathcal{L}_{\mathtt{corr}} &\leq \mathcal{L}_{\mathtt{CORAL}} + \mathcal{L}_{\mathtt{mean}}, \\
\text{where } \mathcal{L}_{\mathtt{CORAL}} &= \left\| \mathbb{E}\left[\left(\hat{\mathbf{h}}^{s} - \mathbb{E}\left[\hat{\mathbf{h}}^{s}\right]\right)\left(\hat{\mathbf{h}}^{s} - \mathbb{E}\left[\hat{\mathbf{h}}^{s}\right]\right)^{\top}\right] - \mathbb{E}\left[\left(\hat{\mathbf{h}}^{\top} - \mathbb{E}\left[\hat{\mathbf{h}}^{\top}\right]\right)\left(\hat{\mathbf{h}}^{\top} - \mathbb{E}\left[\hat{\mathbf{h}}^{\top}\right]\right)^{\top}\right] \right\|_{2} \\
\text{and } \mathcal{L}_{\mathtt{mean}} &= \left\| \mathbb{E}\left[\hat{\mathbf{h}}^{s}\left(\hat{\mathbf{h}}^{s}\right)^{\top}\right] - \mathbb{E}\left[\hat{\mathbf{h}}^{\top}\left(\hat{\mathbf{h}}^{\top}\right)^{\top}\right] \right\|_{2}
\end{aligned}
\tag{27}
$$

Here, $\mathcal{L}_{\mathtt{CORAL}}$ represents the original loss proposed by CORAL (Sun & Saenko, 2016), and $\mathcal{L}_{\mathtt{mean}}$ minimizes the discrepancy between the mean distributions of the source and target domains. Thus, the correlation alignment loss not only aligns the multivariate correlation between the source and target domains, as CORAL does, but also reduces the mean differences between the two domains.

Compared to CORAL and its following works, the correlation alignment loss simultaneously supervises both covariance and mean alignment, ensuring more precise domain alignment. Notably, when the mean distributions of the source and target domains coincide, the correlation alignment loss naturally reduces to the CORAL loss.

Table 5: Comparison of UDA algorithms over the same backbones on the HAR dataset.

| Backbone | Algorithm | 24→27 | 3→13 | 16→13 | 3→8 | 19→2 | 11→28 | 16→10 | 25→10 | 18→10 | 19→10 | AVG |
|---|---|---|---|---|---|---|---|---|---|---|---|---|
| | CORAL | 96.46 | 96.96 | 76.77 | 74.64 | 71.97 | 77.39 | 68.54 | 55.17 | 57.30 | 45.56 | 72.08 |
| Transformer | SASA | 96.46 | 90.90 | 75.75 | 62.35 | 59.34 | 76.52 | 62.92 | 57.30 | 55.05 | 48.31 | 68.49 |
| | CATS | 98.23 | 98.98 | 77.78 | 75.12 | 73.52 | 77.40 | 68.54 | 57.40 | 59.55 | 49.48 | **73.60** |
| | CORAL | 96.46 | 91.91 | 66.67 | 83.53 | 76.81 | 76.52 | 69.66 | 65.17 | 48.31 | 45.79 | 72.08 |
| TimesNet | SASA | 100.00 | 84.84 | 66.67 | 80.00 | 81.32 | 71.30 | 48.31 | 64.04 | 49.34 | 41.57 | 68.74 |
| | CATS | 97.34 | 87.86 | 83.84 | 92.92 | 82.41 | 80.00 | 72.91 | 65.17 | 69.66 | 46.56 | **77.87** |
| | CORAL | 96.46 | 97.97 | 82.82 | 85.86 | 71.42 | 77.39 | 79.20 | 65.17 | 45.50 | 42.94 | 74.47 |
| iTransformer | SASA | 100.00 | 96.97 | 90.90 | 87.06 | 71.42 | 79.13 | 84.27 | 64.41 | 47.68 | 49.43 | 77.13 |
| | CATS | 99.11 | 97.97 | 85.86 | 91.77 | 84.61 | 77.40 | 87.64 | 65.17 | 48.51 | 50.56 | **78.86** |
| | CORAL | 77.00 | 84.84 | 72.72 | 52.94 | 45.05 | 69.56 | 65.16 | 73.03 | 67.41 | 64.04 | 67.18 |
| Crossformer | SASA | 77.87 | 76.76 | 66.67 | 49.41 | 39.56 | 69.56 | 71.91 | 71.91 | 61.80 | 64.05 | 64.95 |
| | CATS | 93.80 | 96.96 | 87.87 | 62.35 | 53.84 | 78.26 | 78.26 | 71.91 | 62.92 | 64.04 | **75.02** |

Table 6: Mitigation rate of CATS across blocks.

| Block | 1 | 2 | 3 |
|---|---|---|---|
| $\gamma$ | 0.6667 | 0.8823 | 0.8359 |

## J   Consistent Comparison on the Same Backbones

To verify that the superior performance of CATS arises from its intrinsic design rather than the capacity of the underlying backbone, we conduct a controlled comparison on the HAR dataset. Specifically, we select two representative UDA baselines (CORAL and SASA) that can be adapted to Transformer-based architectures with minimal modification. We then integrate all three methods (CATS, CORAL, and SASA) with four distinct backbones: Transformer, TimesNet, iTransformer, and Crossformer, and evaluate their performance under identical training configurations.

The experimental results, summarized in Table 5, demonstrate that CATS consistently achieves the best average accuracy across all four backbones, surpassing CORAL and SASA by 4.89% and 6.51%, respectively. This substantial and stable improvement clearly indicates that the effectiveness of CATS does not rely on a specific backbone architecture. Instead, its strong performance stems from its core correlation-aware design, highlighting the method's backbone-agnostic adaptability and its robustness across diverse temporal modeling paradigms.

## K   Empirical Correlation Mitigation

In this section, we design a controlled toy experiment to empirically verify the effectiveness of CATS in mitigating correlation shift. Specifically, we train a three-block CATS-enhanced Transformer on the HAR dataset, using the 24-th domain as the source domain and the 27-th domain as the target domain. After the model converges, we freeze all parameters and extract both the input and output features of each CATS adapter across all layers. To quantify the degree of correlation alignment, we compute the mean of all elements in the corresponding correlation matrices for each sample. Formally, let $\mathbf{H}_{s,i}^{(l)}$ and $\mathbf{H}_{t,i}^{(l)}$ denote the input features at the $l$-th block for the $i$-th test sample from the source and target domains, respectively. Given the CATS adapter $\phi^{(l)}(\cdot)$ in the $l$-th block, we define the set of mean correlation values as

$$
\begin{aligned}
\mathcal{V}_s^{(l)} &:= \{v_{s,i}^{(l)}\}_{i=1}^{N_s}, \quad v_{s,i}^{(l)} = \text{AVG}\big(\text{Corr}(\mathbf{H}_{s,i}^{(l)})\big), \\
\mathcal{V}_t^{(l)} &:= \{v_{t,i}^{(l)}\}_{i=1}^{N_t}, \quad v_{t,i}^{(l)} = \text{AVG}\big(\text{Corr}(\mathbf{H}_{t,i}^{(l)})\big), \\
\tilde{\mathcal{V}}_s^{(l)} &:= \{v_{s,i}^{(l)}\}_{i=1}^{N_s}, \quad v_{s,i}^{(l)} = \text{AVG}\big(\text{Corr}(\phi^{(l)}(\mathbf{H}_{s,i}^{(l)}))\big), \\
\tilde{\mathcal{V}}_t^{(l)} &:= \{v_{t,i}^{(l)}\}_{i=1}^{N_t}, \quad v_{t,i}^{(l)} = \text{AVG}\big(\text{Corr}(\phi^{(l)}(\mathbf{H}_{t,i}^{(l)}))\big),
\end{aligned}
\tag{28}
$$

where $\mathrm{Corr}(\cdot)$ computes the feature-wise correlation matrix and $\mathrm{AVG}(\cdot)$ performs an element-wise average. Here, $N_s$ and $N_t$ denote the number of test samples in the source and target domains, respectively. If no correlation shift exists between the two domains, the corresponding sets of mean correlation values, $\mathcal{V}_s^{(l)}$ ($\tilde{\mathcal{V}}_s^{(l)}$) and $\mathcal{V}_t^{(l)}$ ($\tilde{\mathcal{V}}_t^{(l)}$), should exhibit highly similar distributions. Motivated by this intuition, we measure the magnitude of correlation shift at block $l$ as the absolute difference between the mean correlation values of the two domains:

$$\begin{aligned}
\Delta^{(l)} &= \left| \mathrm{AVG}\big(\mathcal{V}_s^{(l)}\big) - \mathrm{AVG}\big(\mathcal{V}_t^{(l)}\big) \right|, \\
\tilde{\Delta}^{(l)} &= \left| \mathrm{AVG}\big(\tilde{\mathcal{V}}_s^{(l)}\big) - \mathrm{AVG}\big(\tilde{\mathcal{V}}_t^{(l)}\big) \right|,
\end{aligned} \tag{29}$$

A smaller $\Delta^{(l)}$ ($\tilde{\Delta}^{(l)}$) indicates that the representations from the two domains share a more consistent correlation structure.

Furthermore, if CATS successfully mitigates correlation shift, the correlation discrepancy after applying the adapter should be reduced. We therefore define the *mitigation rate* at block $l$ as the ratio

$$\gamma^{(l)} = \frac{\tilde{\Delta}^{(l)}}{\Delta^{(l)}}, \tag{30}$$

where $\gamma^{(l)} < 1$ signifies that the CATS adapter effectively decreases correlation divergence between the source and target domains. Intuitively, a lower mitigation rate corresponds to stronger correlation alignment. Empirically, as shown in Table 6, each CATS adapter consistently mitigates correlation shift across all network blocks, achieving an average mitigation rate of 79.50%. These results provide strong evidence that CATS plays a distinctive and stable role in addressing correlation shift, validating its effectiveness as a correlation-aware domain adaptation mechanism.

## L   More Related Works

**Graph Neural Networks.**   GNNs are effective for capturing dependencies within graphs. Graph Convolutional Networks (GCNs) (Zhang et al., 2019; Kipf & Welling, 2016) aggregate neighbor information by utilizing a localized first-order approximation of spectral graph convolutions. Graph Attention Networks (GATs) (Veličković et al., 2017) implement attention mechanisms that dynamically weigh the contributions of neighboring nodes. GRAND (Feng et al., 2020) learns node representations by randomly dropping nodes to augment data and enforcing the consistency of predictions among augmented data. GraphSAGE (Hamilton et al., 2017) generates embeddings for unseen nodes by sampling and aggregating features from the local neighborhood. For more recent works on GNNs, see (Sharma et al., 2024; Ju et al., 2024; Khoshraftar & An, 2024; Shao et al., 2024).

**Multivariate Time Series Classification.**   Several recent works have sought to advance multivariate time series classification (MTSC) by improving interpretability, efficiency, and adaptability. LAXCAT (Hsieh et al., 2021) employs a CNN with dual attention to simultaneously identify the most informative variables and the temporal intervals that drive predictions, yielding both state-of-the-art accuracy and built-in explainability. DSN (Xiao et al., 2022) uses sparse connections learned via dynamic sparse training to cover multiple scales without extensive hyperparameter tuning. TimeMIL (Chen et al., 2024) formulates classification as a time-aware MIL problem and leverages a time-aware pooling mechanism and a wavelet-positional transformer to better localize sparse, anomalous patterns in long series. LightTS (Campos et al., 2023) compresses ensembles into lightweight student models by learning teacher-specific weights and identifying Pareto-optimal trade-offs.

## M   Limitations

Despite offering a concise and effective solution to the UDA problem in multivariate time series (MTS), our approach still faces two key limitations: (1) CATS is specifically designed under the assumption that there exists a significant correlation shift between domains. If the domain shift is primarily temporal shift, i.e.,

involving only changes in temporal patterns without notable differences in inter-variable correlations, CATS may offer only limited performance gains. (2) In real-world scenarios, time series data often suffer from missing values or irregular sampling. CATS does not incorporate specialized mechanisms to handle such inconsistencies, which may hinder its effectiveness and limit its applicability in these settings.

