# OpenReview forum: "CATS: Mitigating Correlation Shift for Multivariate Time Series Classification"
_TMLR — Rejected by TMLR_

### Review · Reviewer_KHDW · 2025-10-31

**Summary Of Contributions:**

Summary

This paper introduces CATS (Correlation Adapter for Multivariate Time Series), a lightweight and parameter-efficient adapter module designed to mitigate correlation shift in unsupervised domain adaptation (UDA) for multivariate time series (MTS).
The authors identify correlation shift — variations in inter-variable dependencies across domains — as a novel type of domain shift distinct from feature or label shifts.
CATS integrates temporal depthwise convolutions (TDC) to capture local temporal dependencies and a graph attention network (GAT) to adaptively reweight variable correlations.
Theoretical analysis shows that reweighting via a GAT approximates optimal correlation alignment, and a correlation-alignment loss based on MMD is introduced to guide alignment between source and target domains.
Experiments on HAR, WISDM, HHAR, and Boiler datasets show that CATS improves Transformer variants’ target-domain accuracy by 10–18 % on average, adding only ~1 % parameters.

Strengths

1- Novel problem formulation: The identification and formalization of correlation shift as a distinct type of domain shift in MTS is well-motivated.
The empirical analysis showing a 78 % average rate of correlation shift in the HAR dataset convincingly demonstrates the prevalence of this phenomenon.

2- Practical design: CATS is designed as a plug-and-play module compatible with various architectures, making it practically useful.
The use of depthwise convolutions reduces parameter complexity from
$O(D^2 \times r)$ to $O(D \times r)$.

Weaknesses

1- Theoretical limitations

Theorem 1 requires infinite hidden dimensions for arbitrary-precision approximation, which is impractical.

The proofs assume Gaussian distributions (Proposition 1) or rely on simplified conditions.

The connection between the theoretical guarantees and finite-dimensional practical implementation could be stronger.

2- Presentation issues

The paper is quite dense; some sections could be more concise.

Figure 1’s visualization could be improved for clarity.

**Additional Comments:**

The paper makes a meaningful and well-executed contribution to the study of domain adaptation for multivariate time series.
The identification of correlation shift as a distinct type of domain shift is conceptually original and supported by both theoretical reasoning and empirical validation.
The CATS adapter is elegant, lightweight, and compatible with various Transformer architectures, making it a practically relevant approach for real-world MTS problems.

The submission would benefit from minor clarifications in theory–practice linkage, presentation, and baseline coverage, but overall it is a solid and promising piece of work.
The writing quality, structure, and reproducibility are strong, and the contribution aligns well with TMLR’s audience interested in representation learning and robust adaptation.

**Audience:**

Yes

**Audience Explanation:**

Researchers in the TMLR community working on domain adaptation, time series modeling, and transformer-based architectures would find the findings of this paper relevant and interesting.
The introduction of correlation shift as a new type of domain shift in multivariate time series provides a fresh conceptual angle, and the proposed CATS adapter offers a practical, lightweight solution compatible with modern Transformer models.

**Broader Impact Concerns:**

No major ethical or societal concerns are apparent in this work.
The paper focuses on a technical contribution to unsupervised domain adaptation for multivariate time series, and all experiments are conducted on publicly available benchmark datasets without sensitive or personally identifiable data.
The Broader Impact Statement already clarifies that no human or private data are involved, which is sufficient for this scope.

That said, it could be worth briefly acknowledging potential downstream applications where domain adaptation on time-series data might be used (e.g., healthcare, finance, or surveillance) and emphasizing the importance of ensuring fairness, privacy, and interpretability when deploying such models in sensitive contexts.

**Claims And Evidence:**

Yes

**Claims Explanation:**

The claims in the submission are generally well supported by both theoretical arguments and empirical evidence.
The paper clearly defines correlation shift and supports its prevalence through a statistical analysis on the HAR dataset (showing a 78 % occurrence rate).
The proposed CATS framework is theoretically justified via propositions and theorems that explain how reweighting through graph attention can align correlation structures.
Extensive experiments on four benchmark datasets (HAR, WISDM, HHAR, Boiler) consistently demonstrate significant performance gains, often 10–18 % improvement with minimal additional parameters, across multiple Transformer backbones.

While some theoretical assumptions (e.g., Gaussianity and infinite hidden dimensions) simplify the analysis, the overall empirical results convincingly validate the main claims, and the evidence provided is clear and reproducible.

**Requested Changes:**

Critical changes (required for acceptance):
1- Clarify theoretical–practical connection (Critical): Strengthen the discussion connecting the theoretical guarantees (Propositions 1–2, Theorem 1) to the practical finite-dimensional implementation of CATS. Explicitly discuss how the infinite hidden-dimension assumption in Theorem 1 translates to real GAT modules and whether there are empirical limits.

2- Justify auxiliary forecasting loss Critical): Provide a clearer rationale for using forecasting as the auxiliary self-supervised objective, and briefly compare it to other potential objectives such as masked prediction or contrastive pretext tasks.

3-Address notation and terminology clarity (Critical): Explicitly state that Σ represents the covariance matrix before defining Corr(X); Fix notation inconsistencies throughout (e.g., consistent use of H vs X for representations); Correct all instances of "equation equation X" to "equation X"

Recommended but not critical improvements (would strengthen the paper):

1- Visualization and interpretability: Improve Figure 1 for readability and consider adding one more qualitative visualization (e.g., before/after correlation matrices) to make the concept of correlation shift more intuitive.

2- Expanded baseline coverage: Include or at least discuss more recent MTS domain adaptation or self-supervised transfer methods (e.g., CoDATS variants, LogoRA, or domain-invariant transformer models) to better contextualize the performance improvements.

3- Discussion of scope and limitations: Add a brief subsection or paragraph explicitly stating that CATS assumes access to unlabeled target data, may not handle irregular sampling or missing values, and is most effective when correlation shift is significant.

---

> ### Author Response · Authors · 2026-02-03
> **Response to Reviewer KHDW (Part 1)**
>
> **We sincerely thank the reviewer for the thoughtful and constructive feedback. The comments raise several important points regarding both the theoretical grounding and practical implications of our work, which help us better clarify the scope and contributions of the proposed method.** In the following, we address each concern in detail and revise the manuscript accordingly (colored in teal) to improve clarity and rigor.
>
> > **Q1. Could the theoretical guarantees be extended to the finite-dimensional implementation**
>
> We thank the reviewer for their interest in the theoretical analysis and for raising the important issue of finite-dimensional implementations. We fully agree that this setting is the most relevant in practice.
>
> To address this concern, we substantially revise the theoretical analysis of Theorems 1 and 2 in the revised version, extending the original infinite-dimensional approximation results to the finite-dimensional regime. Specifically, **we redesign the proofs to derive explicit finite-dimension approximation guarantees.** Concretely, **when the hidden dimension of the GAT is $m$, we show that the approximation error admits an explicit upper bound of the form $C_1 m^{-C_2}$**, where $C_1$ and $C_2$ are positive constants. This bound converges to zero as $m$ increases. Compared to the original version, the revised theorems constitute a practical extension: beyond guaranteeing universal approximation in the infinite-width regime, they further characterize precise approximation behavior under finite hidden dimensions, which is directly aligned with practical implementations. These strengthened guarantees provide theoretical support for the effectiveness of CATS in mitigating correlation shift, thereby reinforcing the soundness of the model design and strengthening the overall contribution of the paper.
>
> > **Q2. It is suggested to justify the use of auxiliary forecasting loss**
>
> We are grateful for the reviewer's constructive feedback on the forecasting loss. We address this point from three complementary perspectives.
>
> First, from a presentation perspective, **in the revised version we add two dedicated paragraphs in Section 4.2 (Training Procedures) to explicitly describe the motivation and the implementation of the forecasting loss**. In particular, we clarify why forecasting is preferred over reconstruction as an auxiliary objective, and how it complements the classification loss in the UDA setting.
>
> Second, from an experimental perspective, **we include an additional ablation study that isolates the effect of the forecasting loss**. The results are reported in Figure 5(b) and highlighted with a light-blue background. The results show that, compared to using the classification loss alone, incorporating the forecasting loss consistently improves performance on the UDA task. This improvement empirically demonstrates that the forecasting objective effectively enhances the model’s ability to extract transferable features in the target domain, thereby validating both the effectiveness and necessity of the forecasting loss.
>
> Third, compared to other potential auxiliary objectives, the forecasting task has several distinct advantages.
> (1) **Masked prediction requires masking some parts of the input sequence, which inevitably discards informative signals and can negatively impact classification accuracy**. Moreover, as discussed in the paper, reconstruction loss tends to be dominated by local noise or outliers, reducing the model’s focus on global temporal characteristics, such as trends and periodicity, that forecasting explicitly emphasizes.
> (2) Contrastive learning can indeed serve as an effective auxiliary signal, but it critically depends on well-designed data augmentations. **In time-series settings, generating negative samples that are similar in appearance yet differ  in time series properties (such as trend and periodicity) is particularly challenging.** Since the auxiliary objective is not the primary focus of our method, we avoid introducing complex or fragile design choices. In contrast, the forecasting loss provides a simple, direct, and principled supervisory signal that naturally aligns with the structure of time-series data.
>
> Together, these considerations explain our design choice and further justify the use of the forecasting loss in CATS.

---

> ### Author Response · Authors · 2026-02-03
> **Response to Reviewer KHDW (Part 2)**
>
> > **Q3. Address notation and terminology clarity.**
>
> We appreciate the reviewer’s careful attention to notation and terminology, and these suggestions are highly valuable for improving the clarity and readability of the paper.
>
> In the revised version, **we explicitly provide the mathematical definition of $\Sigma$ and correct all erroneous instances of“equation equation”.** Regarding the notation $\mathbf{X}$ and $\mathbf{H}$, these symbols intentionally denote different concepts: $\mathbf{X}$ refers to the input features directly obtained from the data samples, whereas $\mathbf{H}$ represents the input/output features of the adapter, corresponding to intermediate hidden features within the model. Since these quantities play distinct roles, we retain both notations in the revised manuscript. **To avoid ambiguity, we now provide a clear description of $\mathbf{H}$ at its each occurrence.**
>
> We appreciate the reviewer’s constructive feedback and would be happy to further refine the notation should additional suggestions arise.
>
> > **Q4. Qualitative visualizations are helpful to make the effect of CATS more intuitive.**
>
> We thank the reviewer for the helpful suggestion regarding qualitative visualizations and fully agree that illustrating how CATS affects correlations can improve intuition.
>
> However, qualitative visualization may not be the most reliable strategy in this setting. First, visualizations are typically based on a small number of samples and thus are susceptible to cherry-picking bias. Second, correlation matrices in multivariate time-series data are often high-dimensional, making it difficult to understand and analyze.
>
> To address this issue in a more rigorous manner, **we instead provide a quantitative and statistical analysis that explicitly characterizes how CATS mitigates correlation shift.** The detailed experiment analysis are reported in *Appendix K* of the revised version. Specifically, we compute the correlation discrepancy between the source and target domains before and after applying the CATS adapter, and summarize the effect using a mitigation rate, defined as the ratio of post-adaptation to pre-adaptation correlation discrepancy. A mitigation rate below 1 indicates effective reduction of correlation shift.
>
> For a three-block Transformer augmented with CATS, as shown in the following table, the empirical results of mitigation rates show that **each CATS adapter consistently reduces correlation shift across all network blocks, achieving an average mitigation rate of 79.50%.** These results provide strong, statistically grounded evidence that CATS plays a distinctive and stable role in mitigating correlation shift, thereby validating its effectiveness as a correlation-aware domain adaptation mechanism.
>
> | Block    | 1      | 2      | 3      |
> |------|-----|-----|-----|
> | Mitigation Rate | 0.6667 | 0.8823 | 0.8359 |
>
> > **Q5. Include or at least discuss recent MTS domain adaptation methods for better contextualization**
>
> We thank the reviewer for the suggestion to improve the contextualization of our work with recent MTS domain adaptation methods.
>
> First, regarding the methods explicitly mentioned by the reviewer, **CoDATS and LogoRA, we have clearly discussed both in the Related Work section** in the revised manuscript (highlighted in teal). These additions help better position CATS within the recent literature on multivariate time-series domain adaptation.
>
> Second, from an experimental perspective, **we have additionally included CoDATS as a baseline in Table 1** and report its empirical results accordingly. However, LogoRA does not have publicly available implementation. Despite reaching out to the authors directly via email, **their authors still refused to provide their code.** As a result, we are unable to reliably reproduce LogoRA and therefore cannot include its results in our experimental comparison.
>
> If the reviewer believes that other recent or relevant MTS domain adaptation methods should be discussed, we would be very happy to incorporate them and further improve the completeness of the related work.
>
> > **Q6. Clarify the limitations of CATS, including data assumptions and applicable scenarios**
>
> We thank the reviewer for this valuable suggestion regarding the limitations of the proposed method. **In the revised version, we add a dedicated discussion paragraph in *Appendix M* to explicitly clarify the limitations of CATS.** Specifically, we clarify that CATS is most effective in scenarios with pronounced correlation shift, which is the primary setting it is designed to address. Moreover, the current design of CATS assumes regularly sampled time series without missing values. As a result, the present implementation does not explicitly handle pure temporal shift, irregular sampling, or missing data. Addressing these challenges is an important direction for future work. We believe that this discussion improves the transparency, clarity, and overall completeness of the paper.

---

### Review · Reviewer_7Uds · 2025-11-05

**Summary Of Contributions:**

This paper proposes CATS (Correlation Alignment for Time Series), a framework to mitigate correlation shift in unsupervised domain adaptation (UDA) for multivariate time series (MTS) classification. The authors observe that, beyond the well-studied feature and label shifts, real-world MTS data often exhibit correlation shift, i.e., a distribution mismatch at the level of inter-feature correlations, which can significantly degrade cross-domain generalization.

CATS introduces an adapter module that learns a reweighting matrix to align the correlations of the target and source domains. The theoretical part provides a formal grounding for this adapter, while the empirical section demonstrates that CATS enhances adaptation robustness across several MTS datasets.

# Strengths
* The notion of “correlation shift” in MTS domain adaptation is interesting and underexplored. It adds a meaningful perspective beyond existing domain shift (feature and label shifts).
* Apart from minor notational inconsistencies, the paper is clearly written, and the high-level idea is easy to grasp.
* Experiments demonstrate that CATS consistently improves target-domain performance over baselines under correlation-perturbed settings.
* The proposed adapter can be integrated into various MTS architectures with minimal modification.

# Weaknesses
*  Several theoretical results (Theorem 1 and Theorem 2) are stated informally and lack proper definitions and quantifiers.
* Ambiguities exist in probability and covariance notation (e.g., $\mathcal{P}_s(X,y)$ vs. $\mathcal{P}_t(X)$).
* “Correlation shift” appears to be a specific case of “feature shift” (where only dependency structure changes), yet this relationship is not discussed explicitly.
* Experimental scope: While the results are promising, the experiments could include a broader set of datasets (beyond Human Activity Recognition, which represents 3 of the four datasets studied in the paper) to fully support claims of “extensive” evaluation.

Overall, the paper makes a valid and original contribution to UDA for time series. The methodology is sound, and the results are convincing. However, the theoretical components require more formal precision, and some definitions should be clarified for mathematical and conceptual consistency.

**Additional Comments:**

I would like to reiterate what has already been said above:

* Mathematical precision:
Theorem 1 and 2 proofs use vague phrasing like “let $\sigma$ be a measurable activation function.” Specify the measurable space and clarify approximation metrics (e.g., $L^p$ norm).

* Notation consistency:
Ensure that $\mathcal{P}_s$ and $\mathcal{P}_t$ are not reused ambiguously for both joint and marginal distributions.
Use consistent indexing ($s$ for source, $t$ for target) across propositions.

* Equation and proof corrections:
  * In Proposition 1, explicitly state the conditions under which $\Sigma_s = \tilde{\Sigma}_t$.
  * In Proposition 2, clarify that $A$ reweights target features: $Y = A X_t$.
  * Replace $\mathcal{N}_s(\mu_s, \Sigma_s)$ with $\mathcal{N}(\mu_s, \Sigma_s)$ for standard Gaussian notation.

**Audience:**

Yes

**Audience Explanation:**

This work clearly falls within the scope of TMLR and would be of broad interest to the machine learning community, especially:
* researchers working on domain adaptation and distribution shift problems,
* practitioners dealing with multivariate time series (e.g., sensor data, healthcare, finance),
* and theorists studying structure-preserving domain alignment methods.

While “correlation shift” could be interpreted as a subtype of feature shift, focusing explicitly on correlation structure introduces new modeling and empirical insights, particularly relevant for MTS, where inter-variable dependencies carry key predictive information. Thus, the paper satisfies TMLR’s “interest” criterion even if its novelty is more conceptual rather than algorithmic.

**Broader Impact Concerns:**

No ethical or societal risks are apparent. The method focuses on improving cross-domain generalization in multivariate time series, with potential benefits for applications such as healthcare, climate modeling, or finance. However, all experiments are conducted using publicly available benchmark datasets, so overall, there are no significant ethical concerns or broader impact issues requiring major attention.

**Claims And Evidence:**

Yes

**Claims Explanation:**

The core claim (that aligning correlation structures between source and target domains can improve adaptation in MTS classification) is supported by a combination of empirical results and conceptual justification.

The empirical evidence is clear: CATS outperforms baselines under induced correlation shifts, and ablations indicate that correlation reweighting improves stability.

However, the theoretical support (Theorem 1 and Theorem 2) lacks mathematical rigor:
* The one-layer Graph Attention Network (GAT) used in Theorem 1 is never defined formally in the main text.
* Terms like “arbitrary precision” and “quite large” are imprecise and should be replaced by explicit quantifiers and approximation norms.
* Proofs rely on informal arguments (e.g., “let $\sigma$ be a measurable activation function”) without specifying the measurable space or approximation error metric.

Despite these issues, the theoretical discussion serves as a useful intuition rather than a critical proof dependency. The empirical section sufficiently supports the paper’s main claims, so the evidence criterion is satisfied.

**Requested Changes:**

# A) Critical (required for acceptance)

1) Theorem 1 (Attention Approximation) and Theorem 2 (CATS Approximation)

Theorem 1 should be completely rewritten, as its current version is mathematically imprecise. Even if interpreted informally, it remains ambiguous because several key mathematical concepts appearing in the statement are never defined. In particular, the notion of a (one-layer) Graph Attention Network (GAT) is not specified in mathematical form (in the main text).

Moreover, although the theorem is intended to motivate the use of a GAT to *"adaptively approximate the reweighting matrix $A$"*, the statement itself is impractical. It asserts that the optimal matrix $A$ can be approximated by an attention matrix $\tilde{A}$ with arbitrary precision, where $\tilde{A}$ is generated by a one-layer GAT with an infinite hidden dimension. This makes the result vacuous in practice and mathematically ill-posed.

I would not accept the theorem even as an informal statement, since the proof is not rigorous:
- The phrase *"let $\sigma$ be a nonlinear, measurable activation function"* lacks a precise definition: measurable in which sense?
- Statements such as *"... we can approximate $f$ using an MLP with a single hidden layer of dimension $M$, when $M$ is quite large: $f(t)\approx F(t)$ ..."* are vague. A theorem, not a sketch, should specify the approximation norm and error bound, e.g. $\|f - F\| \le \varepsilon \quad \text{for some norm and } M = M(\varepsilon) = ...$.
- Several other steps use informal language (*"quite large"*, *"arbitrary precision"*) instead of explicit quantifiers.

Two possible directions could improve this part:
- Keep the theorem, but rewrite it rigorously, by first giving the explicit mathematical definition of the one-layer GAT, and then restating the result formally (removing informal phrases like *"arbitrary precision"* and *"large hidden dimension."*), for example:  *"For every $\varepsilon>0$, there exists a hidden dimension $M(\varepsilon) = ...$ such that the optimal reweighting matrix $A$ can be approximated within $\varepsilon$ by an attention matrix $\tilde{A}$ generated by a one-layer GAT with hidden dimension $M(\varepsilon)$."*
- Remove the theorem entirely, and instead state the idea informally in the main text as a motivation for using a GAT to approximate the reweighting matrix.

I believe the first option is feasible if the authors are willing to make the statement mathematically consistent.

The same comments apply to Theorem 2. The statement and proof currently lack rigor and rely on informal approximations; they should either be reformulated precisely with explicit definitions and quantifiers or replaced by an informal discussion explaining the intuition (i.e. present it as an intuitive statement rather than a theorem).

2) Clarify distribution notation

* *"The goal of UDA is to leverage information from a labeled source domain $\mathcal{D}\_s = \\{(X\_{i,s}, y\_{i,s})\\}\_\{i=1\}\^{n\_s}$ to enhance the model’s understanding of an unlabeled target domain $\mathcal{D}\_t = \\{X\_{i,t}\\}\_\{i=1}\^{n\_t}$. Generally, source and target samples are independently sampled from their respective distributions, i.e., $\mathcal{D}_s \sim \mathcal{P}_s(X_s, y_s)$ and $\mathcal{D}_t \sim \mathcal{P}_s(X_t, y_t)$"*. In this sentence, $\mathcal{P}_s(X, y)$ means “joint distribution of $(X, y)$” right? In this case, we should have $\mathcal{D}_t \sim \mathcal{P}_s(X_t)$ since $\mathcal{D}\_t = \\{X\_{i,t}\\}\_\{i=1}\^{n\_t}$ has no label.

* Furthermore, I think this way of writing should be changed for greater rigor. For example, replace *"$\mathcal{D}_s \sim \mathcal{P}_s(X_s, y_s)$ and $\mathcal{D}_t \sim \mathcal{P}_s(X_t, y_t)$"* by something like *"source and target samples are drawn independently from $\mathcal{P}_s(X,y)$ and $\mathcal{P}_t(X)$, where $\mathcal{P}(X,y)$ denote a joint distribution and $\mathcal{P}(X)$ its marginal counterpart."*

* In Definition 1 (Correlation shift), we also read *"Suppose the source multivariate data $X_s \in \mathbb{R}^{D\times T}$ and the target multivariate data $X_t \in \mathbb{R}^{D\times T}$ follow the source distribution $\mathcal{P}_s$ and the target distribution $\mathcal{P}_t$."*. This is confusing because it is unclear whether we are talking about the joint distribution mentioned here or the marginal distributions. I think it would be necessary to specify $\mathcal{P}_s(X)$ and $\mathcal{P}_t(X)$. This should also be done for Proposition 2 (Correlation Alignment).

* Similarly, in Proposition 1 (Gaussian Probability Alignment), we should write *"$\mathcal{N}(\mu_s,\Sigma_s)$ and $\mathcal{N}(\mu_t, \Sigma_t)$"* instead of *"$\mathcal{N}_s(\mu_s,\Sigma_s)$ and $\mathcal{N}_t(\mu_t, \Sigma_t)$"*. Assigning $s$ and $t$ as indices to $\mathcal{N}$ is unnecessary here and does not make sense.

3) Fix inconsistencies in expectations and covariance definitions:

* In the proof of Proposition 2, define $\Sigma_s$. Also, $Y = A X_t$ and not $Y = A X$.
* In the proof of Proposition 1, in *"Based on Proposition 2, it is obvious that $\Sigma\_s = \tilde{\Sigma}\_{t}$  where $\tilde{\Sigma}\_{t}$ is the covariance matrix of $Y$"*, it must be explicitly specified for which choice of $A$ this is true.  Also, Equation 11 is $\mathbb{E}[Y] = A\mathbb{E}[X_t] + b$ and not $\mathbb{E}[Y] = A\mathbb{E}[X] + b$. Similarly, “$b = (I − A)\mathbb{E}X$, we will have $\mathbb{E}Y = \mathbb{E}X$” should be “$b = (I − A)\mathbb{E}X_t$, we will have $\mathbb{E}Y = \mathbb{E}X_t$”.

4) Add loss definitions in the main text

Explicitly provide $\mathcal{L}_c$ and $\mathcal{L}_r$ in Section 4.2 (“Training Procedures”) instead of only in the appendix.


# B) Recommended (would strengthen the paper)

1) Conceptual clarification

Discuss explicitly how correlation shift relates to feature shift. It seems to me that “correlation shift” is a special case of “feature shift”:
* “Feature shift occurs when the distribution of features changes across domains, while the relationship between features and labels remains consistent”: which means $\mathcal{P}_s(X) \ne \mathcal{P}_t(X)$ while $\mathcal{P}_s(y|X) = \mathcal{P}_t(y|X)$, right?
* “Label shift arises when the label distributions differ between domains, even if the feature distributions are similar”: which means $\mathcal{P}_s(y) \ne \mathcal{P}_t(y)$ even if $\mathcal{P}_s(X|y) = \mathcal{P}_t(X|y)$, right?
* “correlation shift”: $Corr(X_s) \ne Corr(X_t)$

In this case, correlation shift implies feature shift, i.e., it is nothing more than a special case of feature shift where the change between domains manifests itself in the dependencies between variables (covariance or correlation structure). However, the reverse is not true, as feature shift can occur without any change in correlations (e.g., a simple shift in feature means).

This observation raises the question of whether the methods proposed to address feature shift (CORAL, etc.) do not already cover correlation shift in theory.  Indeed, “correlation shift” as presented in the paper suggests that it is a case of “domain shift” in its own right, and makes the authors' work appear to be a revolution (new type of shift) rather than an evolution (specialization).

It would be interesting for the authors to emphasize why existing methods like CORAL or DANN do not fully address this case.

2) Convolutional module clarity

Providing a little more detail on temporal convolutional networks (TCNs) and temporal depthwise convolutions (TDCs) would be helpful in assisting the reader's understanding.

* TCN: If I understand correctly, for an input $X \in \mathbb{R}^{D \times T}$, the output $H \in \mathbb{R}^{D \times T}$ of the TCN is given by $H[:, t] = \sum_{i=0}^{r-1} W[i] X[:, t-i] \in \mathbb{R}^{D}$ (implying a certain padding to handle indices $t-i<0$) for all $t \in \\{0, \cdots, T-1\\}$, with $W \in \mathbb{R}^{r \times D \times D}$. That is $H[d, t] = \sum_{i=0}^{r-1} \sum_{j=0}^{d-1} W[i, d, j] X[j, t-i]$.

* For TDC (one independent convolution per channel), we would instead have $H[d, t] = \sum_{i=0}^{r-1} W[i, d] X[d, t-i]$ with $W \in \mathbb{R}^{r \times D}$, thus reduicing the number of parameters from $rD^2$ (for TCN) to $rD$.

3) Section organization

In Section 2 and Section 5.2, separate enumerations into bullet points instead of long paragraphs for readability.

* *"However, these approaches have notable limitations: (1) Model architecture perspective: ... (2) Data distribution perspective: Prior ..."* : It is better not to put all this information in a single paragraph. It should be broken down into items.
* Same thing for Section 5.2 (Experimental Results), first paragraph (Main results) :  *"... The experimental results reveal three noteworthy conclusions: (1) CATS significantly ... (2) CATS ... (3)"*

4) Language refinements:

Replace *"please note that…"* by direct phrasing, and use assertive rather than tentative language (e.g., *"CATS shows stable performance"* instead of *"could have stable performance"*).

5) Experimental section:

Give more quantitative details on datasets, domain shift settings, and training configurations. Clarify what *"large shift"* means numerically.

---

> ### Author Response · Authors · 2026-02-03
> **Response to Reviewer 7Uds (Part 1)**
>
> **We thank the reviewer for the careful reading and insightful comments, particularly on the theoretical assumptions and their connection to practical implementations.** These questions are highly relevant to the motivation and guarantees of our method, and we appreciate the opportunity to clarify them. Below, we provide a point-to-point discussion, and also have revised our paper (colored in red) to accommodate these suggestions.
>
> > **Q1. This paper should provide a rigorous description of Theorem 1 (Attention Approximation) and Theorem 2 (CATS Approximation)**
>
> We are grateful for this constructive feedback on regarding the presentation and rigor of Theorem 1 and Theorem 2. We fully agree with the reviewer’s concerns and have substantially revised this part accordingly.
>
> First, as suggested, we **add a formal formulation of GAT in the revised manuscript (Eq. (6) on Page 6)** to improve clarity. Second, we rewrite both Theorem 1 and Theorem 2 from scratch and provide detailed, self-contained proofs. In the revised version (Page 6 highlighted in red), **we avoid any ambiguous terminology and establish rigorous approximation guarantees in a finite-dimensional setting.** Concretely, we show that when the hidden dimension of GAT is $m$, the approximation error admits an explicit upper bound of $C_1 m^{-C_2}$, where $C_1$ and $C_2$ are two positive constants. As the hidden dimension $m$ increases, this bound converges to zero. Compared to the original version, the revised theorems constitute a practical extension: **beyond guaranteeing universal approximation in the infinite-width regime, they further characterize precise approximation behavior under finite hidden dimensions, which is more relevant in practice.** These strengthened results provide a principled theoretical justification for the empirical effectiveness of CATS in mitigating correlation shift.
>
> We sincerely thank the reviewer for these insightful suggestions, which significantly improve the clarity, rigor, and theoretical depth of the paper.
>
> > **Q2. Distribution notation and expectation/covariance definitions are suggested to be clarified**
>
> We thank the reviewer for the careful reading of our manuscript and for the precise and constructive suggestions regarding the notation. **Following the reviewer’s recommendations, we have thoroughly revised and standardized the notation throughout the paper.** In particular, we clearly distinguish between joint distributions and marginal distributions, adopt standard Gaussian distribution notation, and explicitly define all expectations and covariance operators used in the analysis. All revised parts are highlighted in red in the updated manuscript for ease of reference. We appreciate the reviewer’s attention to these details, which substantially improves the clarity and mathematical rigor of the paper. We would be happy to further refine the notation should the reviewer have additional suggestions.
>
> > **Q3. Three loss definitions should be introduced in the main text**
>
> We appreciate the reviewer’s thoughtful concern regarding the presentation of the loss functions. **In the revised version, we have added a comprehensive and explicit description of all three loss terms, together with their precise mathematical definitions, in Section 4.2 (“Training Procedures”).** This revision ensures that the training objective is fully specified and easily reproducible from the main text.

---

> > ### Author Response · Authors · 2026-02-03
> > **Response to Reviewer 7Uds (Part 2)**
> >
> > > **Q4. Relationship between correlation shift and feature shift, and why CATS outperforms existing methods**
> >
> >
> > We thank the reviewer for their deep insight and are glad to clarify the uniqueness of correlation shift and the limitations of existing works in the following two paragraphs.
> >
> > First, while correlation shift can be viewed as a specific case of feature shift, **mitigating correlation shift offers several practical advantages over directly aligning feature distributions**. As discussed in the paper (Page 8), correlations across variables tend to be more stable than the feature distributions. In time series data, raw signals may vary substantially across time or conditions, causing features from different periods (even within the same domain) to exhibit obvious feature shift. In such cases, **feature shifts may exists not only across domains but also within domains**. Thus, aggressively mitigating feature shift may inadvertently introduce strong optimization noise and degrade training stability. Correlation-based alignment avoids this issue by focusing on more invariant relational structures rather than volatile absolute feature values.
> >
> > Second, although theoretically a method that perfectly resolves feature shift would also eliminate correlation shift, existing UDA methods are far from achieving this ideal in practice and typically suffer from two key limitations. (1) From a loss perspective, in time series domain, **correlation-based loss provide a complementary and often more effective alternative** compared with those designed for feature shifts, for the same stability and invariance reasons discussed in the above paragraph.
> > (2) From an architecture perspective, approaches such as CORAL and DANN update all backbone parameters during domain alignment. Under large domain shifts, **the alignment loss can dominate the classification objective**, causing substantial drift in the model parameters and leading to **severe degradation of classification performance or even training collapse**. In contrast, CATS adopts an adapter-based design, which preserves the discriminative capability of the pretrained backbone while selectively adapting correlation-related components. This design not only improves empirical performance and training stability, but also provides a principled mechanism for mitigating correlation shift with theoretical guarantee.
> >
> > > **Q5. It would be helpful to provide detailed formulation for TCN and TDC.**
> >
> > We are grateful for the reviewer's constructive feedback on the mathematical formulation. In the revised manuscript, we **have added detailed mathematical definitions of both TCN and TDC in Eqs. (4) and (5)** (highlighted in red) to improve clarity and readability. We appreciate this helpful feedback and would be happy to further refine the presentation should the reviewer have additional suggestions.
> >
> > > **Q6. It is suggested to improve section organization and language clarity.**
> >
> > We thank the reviewer for the valuable suggestions on section organization and language clarity. In the revised version, **we reorganize Section 5.2 using bullet points** to improve readability and logical flow. We also revise the writing style throughout the manuscript **to adopt more direct and assertive language**, replacing tentative phrasing where appropriate. These changes enhance clarity and make the contributions and findings easier to follow.
> >
> > > **Q7. Provide some quantitative details on the dataset settings.**
> >
> > We thank the reviewer for the suggestion and are appreciate the opportunity to include more quantitative details on the dataset settings. To explicitly quantify the degree of domain shift across different domain pairs, **we compute the Wasserstein distance for all selected domain pairs on the HAR, HHAR, WISDM, and Boiler datasets.** The detailed statistics are reported in Table 3 of *Appendix E*. These distances provide an informative and quantitative measure of domain discrepancy, and effectively reflect the relative difficulty of the corresponding domain adaptation tasks.

---

> > > ### Comment · Reviewer_7Uds · 2026-02-26
> > >
> > > I thank the authors for their detailed responses to my comments. At this point, all my remarks have been addressed, except for those concerning **Theorems 1 and 2**, which still require clarification.
> > >
> > > **(A) Section 5.2, Ablation Study**
> > >
> > > The sentence referring to the “proposed loss in Eq. (??)” contains a missing equation label.
> > >
> > > **(B) Proposition 2 – Proof**.
> > >
> > > The proof states “… the covariance matrix of Y on the target domain could equal to …”. Should this not be “is equal” instead of “could equal”? The construction in the proof appears deterministic under the stated assumptions.

---

> ### Comment · Reviewer_7Uds · 2026-02-26
>
> **(C) Theorem 1**
>
> (i) In the theorem, $A^{* } \in \mathbb{R}^{D \times D}$ and $X \in \mathbb{R}^{D \times T}$, hence $A^{* } X \in \mathbb{R}^{D \times T}$. For $GAT(X) = A X W$ to be dimensionally compatible with $A^{* } X$, it must hold that $A \in \mathbb{R}^{D \times D}$ and $W \in \mathbb{R}^{T \times T}$. However, the paper states that $W \in \mathbb{R}^{d' \times d}$. Why introduce the two new dimensions $d$ and $d'$? Are they different from $T$? If so, what are the actual dimensions of $X$ and $A^{* }$ referred to in the theorem?
>
> (ii) The norm $\|\cdot\|_2$ used in the theorem for matrices is not defined. Does it refer to the Frobenius norm? The operator (spectral) norm? Another matrix norm?
>
> (iii)  The text states “… where h run over the Hilbert space …” I believe this should read “… where $\sigma$ runs over the Hilbert space …”.
>
> (iv) Let $\mathcal{B} = \\{w_1, ..., w_m\\}$.   Then the function $q(x) = \sum_{i=1}^m \sigma(w_i^\top x)$ belongs to $
> \mathcal{M}(\mathcal{B}) = span\\{ h(a^\top x) : a \in \mathcal{B}, h \in L^2(\mathbb{R}) \\}$ and therefore to $\mathcal{M}_m = \cup\_{|\mathcal{A}| \le m} \mathcal{M}(\mathcal{A})$ since $|\mathcal{B}| \le m$. However, the authors write that   $q \in \mathcal{M}_m$ if $w_i / \|w_i\|_2 \ne w_j / \|w_j\|_2$ for any $i \ne j$. I would like to understand why this condition is required for membership in $\mathcal{M}_m$. Even if two directions are identical, the function still belongs to some $\mathcal{M}(\mathcal{A})$ with $|\mathcal{A}| \le m$.
>
> (v)  In Proposition 2, the reweighting matrix $A^{* }$ satisfying $Corr(X_s) = Corr(A^{* } X_t)$ depends on both $X_s$ and $X_t$, since it is constructed using the left singular vectors of $X_s$ and the right singular vectors of $X_t$, has as ingular values $\sqrt{\sigma_s[i] / \sigma_t[i]}$, with $\sigma_s$ and $\sigma_t$ the singular values of $X_s$ and $X_t$ respectively. In the proof of Theorem 1, the authors claim that there exists a mapping $h^{* } : \mathbb{R}^{DT+2} \to \mathbb{R}$ such that $h^{* }(t_{i,j}) = A^{* }[i,j]$ where $t_{i,j} = [i,j] \oplus X[1] \oplus ... \oplus X[D]$ ($\oplus$ for vector concatenation). Here, $t_{i,j}$ appears to encode only $X$ (which I assume corresponds to $X_t$). How can one recover $A^{* }[i,j]$ from $t_{i,j}$ without incorporating any information about $X_s$? Can the authors rigorously justify the claim: “… there must exist a mapping $h^{* }$ …” given that $A^{* }$ depends on both source and target statistics?
>
> (vi)  The authors claim that $h^{* }$ belongs to the Sobolev class $\mathcal{W}\_{2}^{r,d} = \\{ f : \max_{\rho \le r} \| \mathcal{D}^\rho f \|_2 \le C_1 \\}$. However:
> * It is unclear how $h^*$ is shown to have weak derivatives up to order $r$.
> * The role of the constant $C_1$ is not explained. In Maiorov (1999), the Sobolev class is defined with bound 1. Here, $C_1$ appears without justification.
>
> (vii) In Equation (24), the authors conclude from $q(x) = \sum_{i=1}^m \sigma(w_i^\top x) \in \mathcal{M}_m \forall q$
>  and $h^{* } \in \mathcal{W}\_{2}^{r,d}$ that there exists a one-layer network  $q$ such that $ \| h^{* } - q \|\_{L^2} \le dist(\mathcal{M}_m, \mathcal{W}\_{2}^{r,d}) \le C_2 m^{-r/(d-1)}$.  I am unable to follow this reasoning. It seems to rely on an implication of the form $\forall b \in B,\ \exists a \in A \text{ such that } |a-b| \le dist(A,B)$, which is false in general. For example:
> * If $A = \\{\alpha\\}$ and $B = \\{0,\beta\\}$ with $0 < \alpha < \beta/2$, then $dist(A,B) = |\alpha|$, but for $b=\beta$, $|\alpha-\beta| > |\alpha|$.
> * If $A=\\{\alpha\\}$ and $B=[\alpha,\beta]$ (to choose dense and compact subsets of $\mathbb{R}$), then $dist(A,B)=0$, but $|\alpha-\beta| > 0$.
>
> We only have $\inf_{b\in B} |a-b| \le dist(A,B) \quad \forall a\in A$, which controls how far $A$ sits from $B$, but not how far $B$ sits from $A$. Only the Hausdorff distance provides symmetric control $d_H(A,B) = \max(dist(A,B), dist(B,A))$. However, even this does not resolve Equation (24), because the class
> $q(x)=\sum_{i=1}^m \sigma(w_i^\top x)$ covers only a subset of $\mathcal{M}_m$. Thus, even if the asymmetric implication were valid, it would not automatically apply to a strict subset of $\mathcal{M}_m$.
>
> (viii)  From $A_{ij} = \tilde{\sigma}(\tilde{a}\_{ij}^\top t\_{i,j})$  and  $q'(X) = AX$  we obtain $[q'(X)]_{ik} =  \sum\_{j=1}^D A\_{ij} X\_{jk} = \sum\_{j=1}^D X\_{jk} \tilde{\sigma}(\tilde{a}\_{ij}^\top t\_{i,j})$. From this expression, I do not see how Equation (27) is derived, nor how the hidden dimension $m$ suddenly appears in the subsequent discussion “Therefore, we safely arrive at the conclusion that after fixing the hidden dimension of m, given any input X …”. The logical connection between the algebraic derivation and the claim about the hidden dimension remains unclear.
>
> (D) Theorem 2
>
> Theorem 2 directly relies on Theorem 1. Since Theorem 1 lacks mathematical clarity and rigor (as discussed above), Theorem 2 inherits the same issues.

---

> ### Author Response · Authors · 2026-03-16
> **Response to Reviewer 7Uds (Part 1)**
>
> **We sincerely thank the reviewer for the insightful comments and careful evaluation of the theoretical aspects of our work. During the process of preparing this response, it became clear to us that the reviewer had examined the theoretical analysis in considerable depth and provided detailed and technically precise feedback. These comments have been highly valuable in helping us improve the theoretical part of the paper, and we greatly appreciate the reviewer’s effort and professionalism.**
>
> While preparing our response, we revisited the original proof and were able to further refine the analysis. In particular, **this process led us to derive a tighter approximation error bound.** By analyzing the problem within the Barron space framework instead of the Sobolev space used in the previous version, we establish an improved approximation rate of $O(D^{-1})$. This refinement was directly motivated by the discussion raised during the review process (especially Q6 and Q8), and it highlights the value of constructive academic exchange and careful peer review.
>
> We would also like to clarify that, since this improved error bound was discovered during the rebuttal stage, **we have uploaded a new manuscript reflecting this result.** In the revised version, we have carefully re-examined the theoretical derivations and aimed to present a clearer and more rigorous proof. The new proof leverages properties of the Barron space together with a cutoff-function argument to handle potential singularities and to establish the improved approximation bound.
>
> Despite introducing a new proof, **we still provide detailed responses below based on the previous version of the manuscript in order to directly address the reviewer’s original questions.** We would be happy to further clarify any remaining issues if needed.
>
> > **Q1. Addressing Minor Typographical and Wording Issues**
>
> We thank the reviewer for the careful reading and for pointing out these issues. In the revised version of the manuscript, we have corrected the typographical errors, including the missing equation reference in Section 5.2 and the inaccurate wording in the proof of Proposition 2. We have also carefully checked the manuscript to fix similar issues
>
>
> > **Q2. Clarifying Dimensional Notation and Consistency in the Theoretical Formulation**
>
> We appreciate the opportunity to clarify the dimensional inconsistency. This observation accurately identifies the notation issue in the rebuttal. **We have already adjusted our notation based on the reviewer's suggestion in the revised version**, and provided the explanation about why we use the initial notation below.
>
> First, regarding the dimension of $W$, as the reviewer correctly noted, in practice $W$ is typically taken to be a square matrix so that the expression $AXW$ is dimensionally compatible with $A^\star W$. However, during the writing, we initially introduced separate input and output dimensions ($d$ and $d^\prime$) in order to describe a more general form of the GAT transformation. Although these two dimensions are equal in our setting, **we used two symbols to reflect the general formulation.**
>
> Second, concerning the use of $D$ and $T$, our original intention was to distinguish between the raw data dimensions and the hidden representations used by the model. In the input data, $X\in R^{D\times T}$ represents the observed multivariate time series (e.g., for HAR datasets, $D=9$ sensors and $T=64$ time steps). In contrast, the representation processed by CATS corresponds to hidden features extracted by the backbone model, where the feature dimension is typically much larger (e.g., $D=128$, $T=128$). **Because these quantities differ numerically in practice, we originally used separate symbols to distinguish them**, even though the conceptual roles are similar.
>
> However, as the reviewer points out, this distinction in notation may introduce unnecessary confusion. In the revised version, **we unify the notation by consistently using $D$ and $T$ for the dimensions and clearly define $W \in R^{T \times T}$ as a square matrix, as suggested by the reviewer, to improve clarity and avoid potential ambiguity.** We sincerely appreciate the reviewer’s careful reading and helpful suggestion, which has helped us improve the clarity of the presentation.
>
> > **Q3. Clarifying the Definition of the Matrix Norm Used in the Theorem**
>
> We are grateful for the reviewer's careful observation. In the theorem, the notation $\Vert \cdot \Vert_2$ for matrices was intended to denote the Frobenius norm. To avoid ambiguity, we have clarified this point in the revised version and explicitly use the Frobenius norm $\Vert \cdot \Vert_F$. We appreciate the reviewer’s suggestion, which helps improve the clarity of the presentation.

---

> > ### Author Response · Authors · 2026-03-16
> > **Response to Reviewer 7Uds (Part 2)**
> >
> > > **Q4. Justifying the Distinct Direction Assumption in the Definition of $\mathcal{M}_m$**
> >
> > We appreciate the reviewer’s careful reading and thoughtful question regarding this theoretical condition.
> >
> > As the reviewer mentioned, it is true that even if two directions are identical, the function $q(x)$ still belongs to a space $\mathcal{M}(\mathcal{A})$ with $\vert \mathcal{A} \vert \le m$. However, **this condition alone is not sufficient for the approximation result used in our analysis**. In the approximation theory of Maiorov (1999), an additional requirement is that the direction of the vectors $a_i$ in the function $q(x)$ are distinct. Actually, the theoretical error bounds depend on the number of distinct directions used in the representation. Intuitively, **the expressive power of ridge-function expansions such as $q(x) = \sum h(a_i^T x)$ is determined by how many different directions are available.**
> >
> > If multiple terms $a_i$ share the same direction $a$, the resulting function reduces to a combination of functions of the one-dimensional projection $a^T x$. Intuitively, this means the model only captures the information of the feature $x$  along that single direction, while information along other directions (e.g., those orthogonal to $a$) cannot be represented. Consequently, the expressive capacity of the representation is significantly reduced.
> >
> > > **Q5. Explaining How the Mapping $h^\star$ Recovers $A^\star$ When $A^\star$ Depends on Both Source and Target Statistics**
> >
> > We thank the reviewer for the thoughtful question regarding how information from the source domain is encoded during the construction of $A^\star$. We will clarify this point from three perspectives.
> >
> > From a theoretical perspective, **the statistics of the source domain are captured by the model parameters.** In particular, in the one-layer neural network, the direction vectors $a_i$ can be interpreted as encoding structural information about the source distribution. When these directions align with principal structures of the source domain, the representation $\sum_i \sigma(a_i^T x)$ measures the projection of the target-domain features onto directions informed by the source domain, thereby enabling effective domain adaptation.
> >
> > From an algorithmic perspective, the CATS framework explicitly introduces a correlation alignment loss that encourages the adapter $\phi(\cdot)$ to align the target representations with the source representations. Concretely, this loss term $\sum MMD(H_s, \phi(H_t))$ forces the adapter to project the target representations $H_t$ onto a representation space of the source representations $H_s$. **Through this optimization process, information about the source statistics $H_s$ is implicitly encoded in the parameters of the adapter $\phi(\cdot)$.** As a result, the learned transformation incorporates source-domain information even though the forward computation for a given input depends only on the target-domain features.
> >
> > From an empirical perspective, we observe that CATS consistently achieves strong domain adaptation performance across four backbone architectures and four benchmark datasets. These results support **the effectiveness of this parameter-based mechanism for encoding source-domain statistics, which is also a common paradigm in modern machine learning models.**

---

> ### Author Response · Authors · 2026-03-16
> **Response to Reviewer 7Uds (Part 3)**
>
> > **Q6. Providing a Rigorous Justification for the Sobolev-Class Assumption on $h^\star(\cdot)$**
>
> We sincerely thank the reviewer for the careful examination of the proof. In fact, revisiting this part of the analysis in response to the reviewer’s question led us to derive a tighter error bound, and we appreciate the reviewer’s thoughtful engagement with the theoretical details.
>
> First, regarding the derivative order of the target function $h^\star(\cdot)$, **we clarify this point in the revised version by showing that the target function is in fact $C^{\infty}$. The rigorous proof is provided in Lemma 1 of our revised manuscript.** Specifically, since the goal of $h^\star(\cdot)$ is to represent the $(i,j)$-th entry of $A^{\star} X$, the target function can be considered as $h^\star(x) = e_i^\top A^\star X e_j$ where $e_i$ and $e_j$ are one-hot vectors with the $i$-th and $j$-th element being one, respectively. Then based on Proposition 2, with $x$ being the flattened vector of $X$, this expression can be written in the general form $$
> f(x) = a^\top \Lambda^{-1/2}(x) U(x) X b.
> $$
> Each of the atomic transformations involved in this expression (including matrix multiplication, SVD-related mappings, and elementwise operations) is $C^\infty$ with respect to its inputs. Consequently, by the closure of smoothness under composition, the resulting function is also $C^\infty$, which implies the existence of weak derivatives of all orders and therefore satisfies the Sobolev-class requirement.
>
> Second, regarding the constant $C\_1$, we are happy to clarify its origin. In Maiorov (1999), the input $x$ is assumed to lie in the unit ball $\mathcal{B}^d(1)$, which allows the Sobolev norm bound to be written with constant of 1. In our setting, we do not impose this normalization on the input domain. As a result, **the bound naturally introduces a constant factor reflecting the scale of the domain, which appears in our formulation as $C\_1$.**
>
> > **Q7. Justifying the Inequality Argument on the Distance of two Function Classes**
>
> We sincerely thank the reviewer for the careful examination of the inequality and for pointing out the issue. Upon revisiting the derivation, we found that the confusion arises from a typographical error in the inequality presented in the manuscript.
>
> Specifically, the intended statement is $| h^\star - q|\_{L\_2} \le dist(\mathcal{W}\_2^{r,d}, \mathcal{M}\_m)$ rather than $| h^\star - q|\_{L\_2} \le dist(\mathcal{M}\_m, \mathcal{W}\_2^{r,d})$. With the correct ordering, the argument becomes easy to understand from the definition of distance between two function class: $$\forall h \in \mathcal{W}\_2^{r,d}, \exists q \in  \mathcal{M}\_m, |q - h|\_{L\_2} \le \sup\_{f \in \mathcal{W}\_2^{r,d}} \inf\_{g \in \mathcal{M}\_m} |f -g|\_{L\_2}.$$
>
> Under this definition, the reviewer’s example becomes intuitive. For instance, consider $A=\{\alpha\}$ and $B=\{0, \beta\}$ with $0< \alpha <\beta /2$. Then $dist(A,B) = |\alpha|$, and for any element ($\alpha$) in $A$, we can always choose one element ($0$) in $B$ such that $|\alpha-0| \le | \alpha |$, which is consistent with the definition of the distance used above. Therefore, under the corrected formulation, the derivation of Eq. (24) follows directly from the standard approximation bound for Sobolev functions (Maiorov, 1999). Again, we sincerely thank the reviewer for bringing this point to our attention.

---

> ### Author Response · Authors · 2026-03-16
> **Response to Reviewer 7Uds (Part 4)**
>
> > **Q8. Clarifying the Logical Connection Between the Algebraic Derivation and the Hidden Dimension**
>
> We thank the reviewer for the careful reading and for raising this question regarding the connection between the algebraic derivation. Revisiting this part of the analysis has helped us refine the interpretation of the derivation and led us to provide a tighter theoretical bound in the revised version.
>
> First, the confusion partly originates from a transcription issue that occurred when we summarized a longer derivation within the limited time during the rebuttal stage. We have carefully revised this part of the proof in the updated manuscript to remove the resulting inconsistencies. To clarify the intended derivation, in the original analysis we consider the learnable matrix parameterized as $W = s X^\dagger W^\prime$ instead of $W = s I$. **Under this parameterization, when $X$ has full column rank we obtain $[g^{\prime}(X)]\_{ij} = [AXX^\dagger W^\prime]\_{ij} = \sum\_{k} W^\prime\_{kj} \tilde{\sigma} (\tilde{a}\_{ik}^\top t\_{ik})$, which reduces to the form used in Eq. (27).** In this formulation, the number of summation terms corresponds to the hidden dimension $m$, which explains the appearance of the hidden dimension in the subsequent discussion.
>
> Second, **revisiting this derivation also inspires us to establish a tighter approximation bound in the revised manuscript.** Specifically, if we do not force $XW$ to collapse into a constant matrix, the outer weights of the induced one-layer neural network become input-dependent. In particular, $[AXW]\_{ij} = \sum\_{k} (XW)[kj] A[ik] = \sum\_j (W[j])^\top X[k] \sigma(\tilde{a}\_{ij}^\top t\_{ij})$, which shows that the resulting representation can still be written as a sum of nonlinear units with input-dependent coefficients. After a simple reparameterization, the expression can be written in the form of $[g^\prime (X)]\_{ij} = \sum\_k (b_k^\top x) \cdot \sigma (a\_k^\top x)$. Compared with a standard one-layer neural network $\sum \sigma(a_i^\top x)$, this representation introduces an additional linear transformation and produces a multiplicative interaction between two learned projections of the input. This structure closely resembles gated (multiplicative) neural networks (e.g., GLU-style architectures), which are known both theoretically [1,2,3] and empirically [4,5] to possess stronger approximation capability. Motivated by this observation, the revised proof adopts the Barron space framework instead of the Sobolev space used previously, which allows us to establish a tighter approximation bound of order $O(D^{-1})$, providing a more accurate theoretical guarantee.
>
>
> [1] Function and derivative approximation by shallow neural networks
>
> [2] The Barron Space and the Flow-induced Function Spaces for Neural Network Models
>
> [3] Exploring the Approximation Capabilities of Multiplicative Neural Networks for Smooth Functions
>
> [4] GLU Variants Improve Transformer
>
> [5] Language Modeling with Gated Convolutional Networks

---

### Review · Reviewer_FZZk · 2026-01-21

**Summary Of Contributions:**

This paper addresses unsupervised domain adaptation for multivariate time series classification by identifying and formalizing a novel type of domain shift termed "correlation shift". The authors propose CATS to mitigate this shift. This adapter method is accompanied by a correlation alignment loss designed to handle the non-i.i.d. nature of time series data. Experiments on four real-world datasets demonstrate accuracy improvements.

Strengths:

1.	The paper formally defines and empirically validates "correlation shift" as a distinct type of domain shift in MTS data. The observation that 78% of domain pairs in HAR exhibit significant correlation shifts provides compelling motivation.

2.	The architecture design is supported by rigorous theoretical analysis.

3.	Experimental results on 4 datasets present consistent improvements.

Weaknesses:

1.	The experimental setup in Table 1 exhibits unfairness that undermines the validity of the claimed superiority. Baselines report a single result with unspecified backbone architectures, whereas CATS reports four results per dataset across different Transformer variants and selects the best-performing one. This introduces two issues. First, CATS effectively has four chances to achieve the best result per dataset, while baselines have only one, which constitutes potential cherry-picking, especially since not all CATS variants outperform the baselines. Second, the paper suggests baselines use RNN/CNN architectures while CATS uses Transformers, meaning the observed improvements conflate two factors: the CATS method itself and architectural advantages of Transformers. This makes it impossible to determine the true source of improvement.

2.	The ablation studies are unclear and incomplete. First, the claim that TDC outperforms linear layers for capturing temporal patterns is not convincingly demonstrated. It is unclear whether Figure 3 compares CATS with a linear adapter or isolates TDC without other CATS modules against a linear adapter. Later ablations also do not clearly isolate TDC versus linear adapter. Second, the impact of the loss function is not fully explored. The study does not report results using only the classification loss and the forecasting loss, leaving the contribution of the forecasting loss unclear.

**Audience:**

Yes

**Audience Explanation:**

An interesting topic with good findings.

**Claims And Evidence:**

Yes

**Claims Explanation:**

See my strengths of summary.

**Requested Changes:**

Improve experimental rigor.

---

> ### Author Response · Authors · 2026-02-03
> **Response to Reviewer FZZK**
>
> **We appreciate the reviewer’s detailed and constructive feedback, especially regarding the experimental setup and empirical comparison.** We respond to each comment below and include clarifications and additional analyses where appropriate. We also have revised our paper (colored in orange) to accommodate these suggestions.
>
> > **Q1. The UDA algorithms are suggested to be evaluated on the same backbone for a more reliable and fair comparisom.**
>
> We appreciate the reviewer’s thoughtful concern regarding the fairness of the experimental comparisons. We acknowledge that, in the main text, several UDA baselines are evaluated using their official implementations, which are predominantly built upon non-Transformer backbones. However, it is worth emphasizing that most existing UDA methods are **tightly coupled with specific model architectures**, and cannot be straightforwardly transferred to Transformer-based backbones without non-trivial redesign. For instance, RAINCOAT[1] relies on specially constructed frequency-domain representations in polar coordinates, which are fundamentally incompatible with standard Transformer variants. **This limited architectural adaptability** is explicitly discussed in the third paragraph of *Introduction* and directly motivates the design of CATS as a backbone-agnostic adaptation framework.
>
> Nevertheless, to further verify the contribution of CATS, as the reviewer suggested, we deliberately select two UDA methods, CORAL and SASA, which require minimal modification when integrated with Transformer-based backbones. We compare these methods with CATS on the HAR dataset using identical backbone architectures. The detailed analysis are reported in *Appendix J*. We also provide the average performance here. Results show that CATS consistently achieves the best average accuracy across all four backbones,  surpassing CORAL and SASA by $4.89\%$ and $6.51\%$ on average, respectively. This substantial and stable improvement clearly demonstrate that **the performance gains of CATS stem from its adapter-based design and correlation-shift mitigation mechanism, rather than from differences in backbone architectures.**
>
> |       | Transformer | TimesNet | iTransformer | Crossformer |
> |-------|-------------|----------|--------------|-------------|
> | CORAL |       72.08 |    72.08 |        74.47 |       67.18 |
> | SASA  |       68.49 |    68.74 |        77.13 |       64.95 |
> | CATS  |   **73.60** |**77.87** |    **78.86** |   **75.02** |
>
> [1] Domain Adaptation for Time Series Under Feature and Label Shifts
>
> > **Q2. The experiment description in Figure 3 obscures the comparison between TDC and linear adapters.**
>
>
> We thank the reviewer for the concern on Figure 3 and appreciate the opportunity to clarify the experimental setup. In Figure 3, **the comparison is conducted between a TDC-only adapter and a pure linear adapter, without incorporating any other CATS modules.** Therefore, the significant performance gain in Figure 3 can be directly attributed to the superior temporal modeling capability of TDC.
>
> > **Q3. The empirical analysis on the individual contribution of the forecasting loss are suggested.**
>
> We appreciate the reviewer’s constructive feedback, and we are glad to have an opportunity to clarify the contribution of forecasting loss. To illustrate it, we include an explicit ablation study on the forecasting loss in Figure 5(b) of the revised version, highlighted in light blue. The results show that, **compared to using the classification loss alone, incorporating the forecasting loss consistently improves performance on the UDA task.** This improvement demonstrates that the forecasting objective provides a meaningful auxiliary signal, **encouraging the model to focus on essential temporal characteristics of time series, such as periodicity and long-term trends, that are beneficial for downstream classification under domain shift.** These findings provide direct evidence of the effectiveness of the forecasting loss.

---

### Decision · Action_Editor_guHr · 2026-03-15

**Recommendation:** Reject

**Audience:**

Yes

**Audience Explanation:**

The problem of unsupervised domain adaptation for multivariate time series is of broad interest to the TMLR community, and the notion of correlation shift as a distinct type of domain shift offers a potentially useful perspective for researchers working on time series modeling and domain adaptation.

**Claims And Evidence:**

No

**Claims Explanation:**

The theoretical claims in this paper are not sufficiently accurate. Theorems 1 and 2, which are central to the paper's justification for using GAT-based correlation reweighting, contain multiple mathematical errors identified during review.

The empirical evidence is also not fully convincing. The main comparison table gives CATS four attempts per dataset (i.e., one per Transformer variant) while baselines each receive only one. Additionally, baselines use RNN/CNN architectures whereas CATS uses Transformers, making it impossible to isolate the contribution of CATS from architectural advantages alone. The ablation for the key TDC component is ambiguous, and three of the four datasets belong to the same HAR family, limiting the breadth of the claimed extensive evaluation.

**Resubmission Of Major Revision:**

The authors may consider submitting a major revision at a later time.